# IP$_3$ mediated global Ca$^{2+}$ signals arise through two temporally and spatially distinct modes of Ca$^{2+}$ release

**Jeffrey T Lock[1]\*, Ian Parker[1,2]**

[1]Department of Neurobiology & Behavior, UC Irvine, Irvine, United States;
[2]Department of Physiology & Biophysics, UC Irvine, Irvine, United States

**Abstract** The 'building-block' model of inositol trisphosphate (IP$_3$)-mediated Ca$^{2+}$ liberation posits that cell-wide cytosolic Ca$^{2+}$ signals arise through coordinated activation of localized Ca$^{2+}$ puffs generated by stationary clusters of IP$_3$ receptors (IP$_3$Rs). Here, we revise this hypothesis, applying fluctuation analysis to resolve Ca$^{2+}$ signals otherwise obscured during large Ca$^{2+}$ elevations. We find the rising phase of global Ca$^{2+}$ signals is punctuated by a flurry of puffs, which terminate before the peak by a mechanism involving partial ER Ca$^{2+}$ depletion. The continuing rise in Ca$^{2+}$, and persistence of global signals even when puffs are absent, reveal a second mode of spatiotemporally diffuse Ca$^{2+}$ signaling. Puffs make only small, transient contributions to global Ca$^{2+}$ signals, which are sustained by diffuse release of Ca$^{2+}$ through a functionally distinct process. These two modes of IP$_3$-mediated Ca$^{2+}$ liberation have important implications for downstream signaling, imparting spatial and kinetic specificity to Ca$^{2+}$-dependent effector functions and Ca$^{2+}$ transport.

**\*For correspondence:**
lockj@uci.edu

**Competing interests:** The authors declare that no competing interests exist.

## Introduction

Cytosolic Ca$^{2+}$ signals generated by the liberation of Ca$^{2+}$ ions sequestered in the endoplasmic reticulum (ER) through inositol trisphosphate receptor (IP$_3$R) channels regulate ubiquitous cellular processes as diverse as gene transcription, secretion, mitochondrial energetics, electrical excitability and fertilization (*Clapham, 2007*; *Berridge et al., 2000*). Cells achieve such unique repertoires of Ca$^{2+}$-dependent functions by generating a hierarchy of cytosolic Ca$^{2+}$ signals with markedly different spatial scales and temporal durations, ranging from brief, localized Ca$^{2+}$ transients called puffs (*Parker and Yao, 1991*; *Yao et al., 1995*) to larger and more prolonged Ca$^{2+}$ elevations that engulf the cell. Global elevations in cytosolic Ca$^{2+}$ typically last several seconds and may appear as waves that propagate throughout the cell (*Woods et al., 1986*). They can recur as oscillations with periods between a few seconds and a few minutes, and are thought to encode information in a 'digital' manner, whereby increasing stimulus strength results predominantly in an increase in frequency rather than amplitude (*Parekh, 2011*; *Smedler and Uhlén, 2014*). Puffs, on the other hand, are tightly localized elevations in cytosolic Ca$^{2+}$ generated by stationary clusters containing small numbers of IP$_3$Rs, which last only tens or a few hundreds of milliseconds and remain restricted within a few micrometers (*Bootman et al., 1997a*; *Parker et al., 1996*).

The patterning of cellular Ca$^{2+}$ signals evoked by IP$_3$ is largely determined by the functional properties of the IP$_3$Rs and by their spatial arrangement in the ER membrane. Crucially, the opening of IP$_3$R channels is regulated by cytosolic Ca$^{2+}$ itself, in addition to IP$_3$. Low concentrations of Ca$^{2+}$ increase the open probability of the channel whereas high concentrations favor a closed state (*Parker and Ivorra, 1990*; *Iino, 1990*; *Bezprozvanny et al., 1991*). This biphasic modulation of IP$_3$Rs by Ca$^{2+}$ leads to the phenomenon of Ca$^{2+}$-induced Ca$^{2+}$ release (CICR). Ca$^{2+}$ diffusing from one open channel may thus trigger the opening of adjacent channels, with self-reinforcing CICR

countered by $Ca^{2+}$ feedback inhibition. The clustered distribution of IP$_3$Rs further shapes the extent of this regenerative process. CICR may remain locally restricted to a single cluster, containing from a few to a few tens of functional IP$_3$Rs, to produce a puff; whereas it is proposed that a global response is generated by successive cycles of CICR and $Ca^{2+}$ diffusion acting over longer spatial ranges to recruit successive puff sites (*Bootman et al., 1997a*; *Parker et al., 1996*; *Berridge, 1997*; *Bootman and Berridge, 1996*; *Callamaras et al., 1998*; *Dawson et al., 1999*; *Marchant, 2001*; *Marchant et al., 1999*). However, the transition between these modes remains an area of active investigation (*Rückl and Rüdiger, 2016*; *Miyamoto and Mikoshiba, 2019*; *Sneyd et al., 2017a*; *Sneyd et al., 2017b*; *Rückl et al., 2015*), and recent theoretical simulations have questioned whether $Ca^{2+}$ released through puff activity is alone sufficient to propagate global cytosolic $Ca^{2+}$ signals (*Piegari et al., 2019*).

Here, we examined the nature of $Ca^{2+}$ liberation through IP$_3$Rs during global cellular $Ca^{2+}$ signals, asking whether this accords with the widely-accepted 'building block' model (*Bootman et al., 1997a*; *Parker et al., 1996*; *Berridge, 1997*; *Bootman and Berridge, 1996*; *Marchant, 2001*; *Marchant et al., 1999*; *Mataragka and Taylor, 2018*) in which global signals are constructed by the summation of coordinated, pulsatile activation of $Ca^{2+}$ release at puff sites; or whether global signals also involve an additional mode of $Ca^{2+}$ liberation that is more homogeneous in space and time. Although $Ca^{2+}$ puffs are often evident during the initial rising phase of global $Ca^{2+}$ signals (*Bootman et al., 1997a*; *Bootman and Berridge, 1996*; *Marchant, 2001*; *Marchant et al., 1999*), a challenge in answering this question arises because puffs become obscured as the overall cytosolic $Ca^{2+}$ level continues to increase. To reveal and monitor temporally rapid and spatially confined $Ca^{2+}$ transients (puffs) during even large amplitude global $Ca^{2+}$ elevations we developed image processing and analysis routines to analyze local fluctuations in $Ca^{2+}$ fluorescence signals (*Ellefsen et al., 2019*). We applied these routines to $Ca^{2+}$ recordings obtained both by total internal reflection fluorescence (TIRF) microscopy to resolve signals arising near the plasma membrane, and by lattice light-sheet (LLS) microscopy to acquire optical sections through the cell interior. We find that rapid flurries of $Ca^{2+}$ puffs accompany the rising phase of global $Ca^{2+}$ signals evoked by photoreleased IP$_3$ and by agonist stimulation of the IP$_3$ signaling pathway, but these rapidly terminate before the peak of the response through a mechanism regulated by ER $Ca^{2+}$ store content. The punctate liberation of $Ca^{2+}$ via transient, localized $Ca^{2+}$ puffs contributes only a small fraction of the total $Ca^{2+}$ liberated during global $Ca^{2+}$ signals, which are instead sustained by diffuse $Ca^{2+}$ liberation through a functionally distinct mode of release. These two modes of IP$_3$-mediated $Ca^{2+}$ release will likely selectively activate different populations of effectors; those positioned close to the IP$_3$R clusters that mediate puffs and which respond to brief, repetitive transients of $[Ca^{2+}]$, and others that respond to a more sustained, spatially diffuse elevation of bulk cytosolic $[Ca^{2+}]$.

## Results

### Fluctuation processing of $Ca^{2+}$ images highlights transient signals

Our central question was whether IP$_3$-evoked $Ca^{2+}$ liberation during cell-wide $Ca^{2+}$ signals arises through coordinated activation of pulsatile, spatially -localized events, analogous to the local $Ca^{2+}$ puffs observed with weaker IP$_3$ stimulation or after loading cells with EGTA to suppress global signals (*Dargan and Parker, 2003*; *Smith et al., 2009*). To better visualize and identify transient, localized $Ca^{2+}$ events occurring during the course of larger, global elevations of $Ca^{2+}$, we developed an image processing algorithm to highlight and quantify temporal fluctuations of the $Ca^{2+}$ fluorescence signal. We previously described the use of pixel-by-pixel power spectrum analysis of temporal $Ca^{2+}$ fluctuations for this purpose (*Swaminathan et al., 2020*), but this is computationally intensive and unfeasible for large data sets. Here, we adopted a faster approximation, by first temporally band-pass filtering image stacks and then calculating the standard deviation (SD) of the fluorescence fluctuations at each pixel over a running time window (*Ellefsen et al., 2019*).

The conceptual basis of the algorithm is illustrated in *Figure 1* (see also *Figure 1—video 1*). WT HEK293 cells were loaded with the fluorescent $Ca^{2+}$ indicator Cal520 and imaged by TIRF microscopy during global cytosolic $Ca^{2+}$ signals. The panels in *Figure 1A* show Cal520 fluorescence of individual image frames of a HEK cell captured before (i) and after (ii-v) photorelease of i-IP$_3$, an active, metabolically stable analog of IP$_3$. Photoreleased i-IP$_3$ evoked a widespread increase in fluorescence

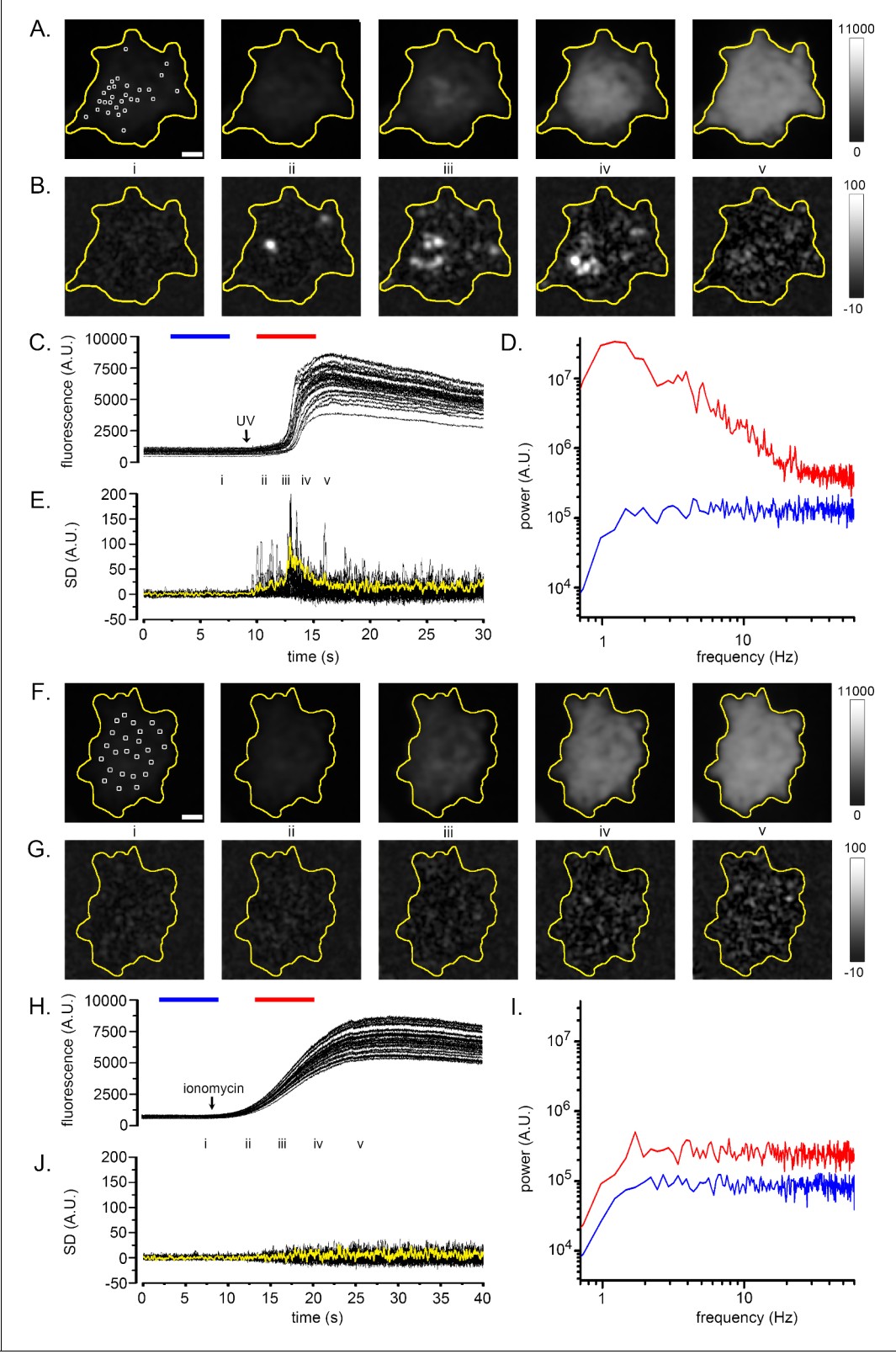

**Figure 1.** Fluctuation analysis of Ca$^{2+}$ signals. (**A–D**) Records from a single WT HEK cell loaded with Cal520 and stimulated by photorelease of i-IP$_3$ to evoke a global Ca$^{2+}$ elevation. (**A**) Panels show 'raw' TIRF fluorescence images of the cell before (i), during the rising phase (ii-iv) and at the peak (v) of the global Ca$^{2+}$ signal. Images are Gaussian-blurred (sigma ~1 µm) single frames (8 ms exposure time) captured at times as marked in C. Grey scale intensities depict fluorescence in arbitrary camera units, as indicated by the bar at the right. The yellow outline marks the TIRF footprint of the cell. (**B**)

*Figure 1 continued on next page*

*Figure 1 continued*

Panels show corresponding standard deviation (SD) images at the same times as in A, highlighting hot spots of local, transient $Ca^{2+}$ release. Grey scale intensities (arbitrary units; A.U.) represent the shot noise-corrected standard deviation of fluorescence fluctuations within a 160 ms running time window. (C) Overlaid black traces show fluorescence monitored from 24 regions of interest (ROIs; marked by squares in panel Ai) placed on areas of local $Ca^{2+}$ activity. The arrow indicates the time of the photolysis flash. (D) Power spectra of $Ca^{2+}$ fluorescence fluctuations averaged from the 24 ROIs at baseline (blue trace) and during the rising phase of the global $Ca^{2+}$ signal (red trace). Spectra were calculated from recordings during the respective times indicated by the colored bars in C, after low-pass (1 Hz) filtering of the fluorescence image stack to strip out the slow rise of the global signal. (E) Overlaid traces show shot noise-corrected SD signals from the 24 ROIs centered on hot spots of $Ca^{2+}$ activity. The thicker yellow trace shows the mean SD signal monitored from a ROI encompassing the entire cell and is depicted after scaling up by a factor of 10 relative to the traces from small ROIs. (F–I) Corresponding images and plots from an HEK cell devoid of $IP_3Rs$ (3KO) in which a global $Ca^{2+}$ elevation was evoked by pipetting a 10 μl aliquot of ionomycin into the bathing solution at a distance from the cell when marked by the arrow in H. In this case no hot spots or increased low-frequency fluctuations accompanied the elevation in cytosolic $[Ca^{2+}]$, and the ROIs (marked by squares in panel Fi) used to derive the data in H-J were placed randomly. The yellow trace in J depicting the mean SD signal from the entire cell is scaled up by a factor of 10 relative to the traces from small ROIs. Fluorescence and SD magnitudes are expressed in arbitrary units consistent with those in A-D.

The online version of this article includes the following video and figure supplement(s) for figure 1:

**Figure supplement 1.** Optimization of space-time parameters used in SD fluctuation analysis algorithm.

**Figure supplement 2.** Correction of signal variance for photon shot noise.

**Figure supplement 3.** Spatial fluctuation analysis of $IP_3$-mediated global $Ca^{2+}$ signals mirrors temporal fluctuations.

**Figure 1—video 1.** Fluctuation processing of $Ca^{2+}$ image recordings.

https://elifesciences.org/articles/55008#fig1video1

---

throughout the cell that peaked within about 5 s, during which time several transient, local 'hot spots' were evident. These are visible in *Figure 1—video 1* but are not readily apparent in *Figure 1A* because of the extended grey scale required to encompass the peak global fluorescence signal. To illustrate the activity at local hot spots, we monitored fluorescence from regions of interest (ROIs) centered on 24 sites (*Figure 1C*). Traces from these sites showed progressive, large fluorescence increases above the baseline, with small, superimposed transients (puffs) during the rising phase. To better discriminate these localized signals, we high-pass (1 Hz) filtered the image stack, pixel-by-pixel, to strip out the slow increase in global fluorescence. *Figure 1D* shows mean power spectra averaged from the 24 sites during image sequences (5 s) acquired before (control; blue trace) and immediately following (red trace) photorelease of i-$IP_3$. The control, baseline spectrum showed substantially uniform power across all frequencies above the applied 1 Hz high-pass filter, compatible with the dominant noise source arising from 'white' photon shot noise. Strikingly, the spectrum obtained during the rise of the global $Ca^{2+}$ signal showed much greater power at frequencies between about 1–20 Hz as compared to the control spectrum, rolling off at higher frequencies to a noise floor determined by photon shot noise. We thus developed an approach to isolate the low-frequency fluctuations attributable to transient $Ca^{2+}$ puffs, while subtracting the photon shot noise that would arise in linear proportion to the overall fluorescence intensity.

Beginning with a black level-subtracted 'raw' fluorescence image stack, our algorithm applied a spatial filter (Gaussian blur with sigma ~1 μm), and a band-pass temporal Butterworth filter (3–20 Hz). The resulting image stack was then processed by a running boxcar window (160 ms) that, for each pixel, calculated the standard deviation (SD) of the fluorescence signal at that pixel throughout the duration of the window. These parameters were chosen to optimally 'tune' the algorithm to reject slow changes in baseline fluorescence and attenuate high-frequency photon shot noise while retaining frequencies resulting from puff activity (*Figure 1—figure supplement 1*). Lastly, the algorithm corrected for photon shot noise by subtracting a scaled measure of the square root of fluorescence intensity at each pixel. If measurements were in terms of numbers of detected photons, the SD would equal the square root of the intensity; however, that was not the case for our records because of considerations including the camera conversion factor and the filtering applied to the image stack. We thus empirically determined an appropriate scaling factor, by determining the linear slope of a plot of mean variance *vs.* mean fluorescence emission from a sample of fluorescein where photon shot noise was expected to be the major noise source (*Figure 1—figure supplement 2*).

*Figure 1B* presents representative SD images calculated by the algorithm, at time points corresponding to the panels in *Figure 1A*, and *Figure 1—video 1* shows fluorescence and SD images throughout the response. The SD signal was uniformly close to zero throughout the cell before

stimulation (*Figure 1B*, panel i), while discrete, transient hot spots were clearly evident at several different sites during the rising phase of the global $Ca^{2+}$ elevation (panels ii-iv), but ceased at the time of the peak response (panel v). This behavior is further illustrated by the black traces in *Figure 1E*, showing overlaid SD measurements from the 24 hot spots of activity. A flurry of transient events at these sites peaked during the rising phase of the global $Ca^{2+}$ response to photoreleased i-IP$_3$ but had largely subsided by the time of the maximal global $Ca^{2+}$ elevation. Even though the global $Ca^{2+}$ level then stayed elevated for many seconds the mean SD signals at these regions remained low. Measurement of the SD signal derived from a ROI encompassing the entire cell (yellow trace, *Figure 1E*) closely tracked the aggregate kinetics of the individual puff sites.

To further validate the fluctuation analysis algorithm, we examined a situation where cytosolic $[Ca^{2+}]$ was expected to rise in a smoothly graded manner, without overt temporal fluctuations or spatial heterogeneities. For this, we imaged Cal520 fluorescence by TIRF microscopy in HEK293 3KO cells in which all IP$_3$R isoforms were knocked out (*Alzayady et al., 2016*). We pipetted an aliquot of ionomycin (10 µl of 10 µM) into the 2.5 ml volume of $Ca^{2+}$-free bathing solution at a distance from the cell chosen so that the diffusion of ionomycin evoked a slow liberation of $Ca^{2+}$ from intracellular stores to give a fluorescence signal of similar amplitude (8.3 $\Delta F/F_0$) and kinetics to that evoked by photoreleased i-IP$_3$ (6.9 $\Delta F/F_0$) in *Figure 1A,C*. *Figure 1F* shows snapshots of 'raw' fluorescence captured before (i) and during (ii-v) application of ionomycin. The fluorescence rose uniformly throughout the cell without any evident hot spots of local transients in the SD images (*Figure 1G* and *Figure 1—video 1*). Measurements from 24 randomly located ROIs (squares in *Figure 1F*) showed only smooth rises in fluorescence (*Figure 1H*). Mean spectra from these regions (*Figure 1I*) displayed flat, substantially uniform distributions of power across all frequencies, consistent with photon shot noise increasing in proportion to the mean fluorescence level. Notably, SD signals from local ROIs (*Figure 1J*, superimposed black traces) and from a ROI encompassing the entire cell (yellow trace) showed no increase in fluctuations beyond that expected for photon shot noise.

## Temporal fluctuations reflect spatially localized $Ca^{2+}$ signals

The SD image stacks generated by the temporal fluctuation algorithm showed transient hot spots of $Ca^{2+}$ release associated with temporal fluctuations. However, the SD signal could also include temporal fluctuations in fluorescence that were spatially blurred or uniform across the cell. To determine whether these contribute appreciably, or whether the SD signal could be taken as a good reporter of localized puff activity, we developed a second algorithm to reveal spatial $Ca^{2+}$ variations in Cal520 fluorescence image stacks (*Figure 1—figure supplement 3*).

$Ca^{2+}$ image stacks were first temporally bandpass filtered as described above. The algorithm then calculated, frame by frame, the difference between strong and weak Gaussian blur functions (respective standard deviations of about 4 and 1 µm at the specimen), essentially acting as a spatial bandpass filter to attenuate high spatial frequencies caused by pixel-to-pixel shot noise variations and low-frequency variations resulting from the spread of $Ca^{2+}$ waves across the cell, while retaining spatial frequencies corresponding to the spread of local $Ca^{2+}$ puffs. The resulting spatial SD images were remarkably similar to images generated by the temporal fluctuation analysis routine (*Figure 1—figure supplement 3A*), and traces of mean cell-wide temporal and spatial SD signals during $Ca^{2+}$ elevations matched closely (*Figure 1—figure supplement 3B–E*). We thus conclude that the temporal SD signals faithfully reflect transient, localized $Ca^{2+}$ puff activity while minimizing confounding contributions from shot noise and slower changes in global fluorescence.

## Fluctuation analysis reveals a transient flurry of puffs during global $Ca^{2+}$ signals

In *Figure 1*, we show traces from discrete subcellular regions to illustrate how temporal SD images detect transient, local $Ca^{2+}$ elevations while being insensitive to homogeneous global $Ca^{2+}$ elevations. However, for all the following experiments in this paper we show SD signals derived from single ROIs that completely encompassed each cell, so as to obtain an aggregate measure of puff activity throughout the cell and obviate any subjective bias that might arise in selecting smaller, subcellular regions. Unless otherwise stated, all imaging was done by TIRF microscopy with cells bathed

in a zero $Ca^{2+}$ solution including 300 µM EGTA to avoid possible complication from entry of extra-cellular $Ca^{2+}$ into the cytosol.

*Figure 2* and *Figure 2—videos 1* and *2* present records from WT HEK cells loaded with Cal520 and caged i-IP$_3$ showing how the SD signal reveals the patterns of puff activity underlying global $Ca^{2+}$ signals. Under basal conditions, the shot noise-corrected cell-wide SD signals were almost flat, with a mean around zero (*Figure 2A*, *Figure 2—video 1*), indicating a negligible level of local $Ca^{2+}$ activity at rest. Photorelease of small amounts of i-IP$_3$ by brief (~100 ms) UV flashes evoked $Ca^{2+}$ puffs - directly visible in the Cal520 fluorescence ratio movie in *Figure 2—video 1*, and more evident as sharp transients in the whole-cell SD trace - but without generating any appreciable global rise in basal $Ca^{2+}$ (*Figure 2B*). Longer flashes (200–1000 ms) generated whole-cell elevations in cytosolic $Ca^{2+}$ that rose and fell over several seconds, with fluorescence signals reaching peak amplitudes in rough proportion to the flash duration (smooth traces, *Figure 2C–E*; *Figure 2—video 2*). SD movies (*Figure 2—video 2*) and whole-cell SD traces (noisy traces, *Figure 2C–E*) revealed an underlying flurry of localized, transient $Ca^{2+}$ events during the rising phase of the global $Ca^{2+}$ responses. In instances where global $Ca^{2+}$ signals were small and slowly rising, the SD traces showed $Ca^{2+}$ transients persisting throughout the prolonged rising phase (*Figure 2C*). On the other hand, the SD traces from cells exhibiting intermediate (*Figure 2D*) and fast rising (*Figure 2E*) global signals revealed $Ca^{2+}$ fluctuations that began almost immediately following photorelease of i-IP$_3$, reached a maximum during the rising phase of the global signal, but then declined almost to baseline by the peak of the response.

The records in *Figure 2A–E* and *Figure 2—videos 1* and *2* illustrate representative responses in individual cells. To pool data from multiple cells we grouped records into categories matching the examples in *Figure 2B–E*: that is responses showing puffs without an appreciable elevation of global $Ca^{2+}$; and slow-, intermediate- and fast- rising global $Ca^{2+}$ responses. *Figure 2G* shows overlaid traces depicting the mean Cal520 fluorescence ratios ($\Delta F/F_0$) of the global $Ca^{2+}$ responses from cells in these different categories, and *Figure 2H* shows the associated mean SD traces. Notably, in all three categories where global $Ca^{2+}$ signals were evoked (*Figure 2C–E*) the mean SD signals were transient, indicating that puff activity was largely confined to the rising phase of the global $Ca^{2+}$ elevation and largely ceased by the time the global signal reached a maximum. The durations of the puff flurries progressively shortened with increasing rates of rise in global $Ca^{2+}$ and the magnitudes of the SD signal at the peak of the flurry activity increased.

## $Ca^{2+}$ signals evoked by agonist activation and photoreleased i-IP$_3$ show similar patterns of puff activity

UV photorelease of i-IP$_3$ provides a convenient tool to activate IP$_3$Rs with precise timing and control of the amount released. However, this IP$_3$ analog is slowly metabolized by the cell, remaining elevated for minutes following photo-uncaging (*Smith et al., 2009*; *Dakin and Li, 2007*), and its uniform release throughout the cell differs from endogenous generation of IP$_3$ at the cell membrane (*Keebler and Taylor, 2017*; *Lock et al., 2017*). We thus compared responses evoked by photoreleased i-IP$_3$ with those activated by the G-protein coupled muscarinic receptor agonist carbachol (CCH), locally applied through a picospritzer-driven micropipette (puffer pipet) positioned above WT HEK cells bathed in zero $Ca^{2+}$ medium. A brief (5 s) pulse of CCH elicited a rapid, global rise in $Ca^{2+}$ that was accompanied by an underlying burst of local $Ca^{2+}$ signals (*Figure 2F*; *Figure 2—video 3*). As with responses evoked by photoreleased i-IP$_3$, fluctuations arising from local $Ca^{2+}$ signals occurred predominantly during the initial portion of the rising phase and then subsided to near basal levels before the peak of the global response. *Figure 2I* shows mean traces of whole-cell global $Ca^{2+}$ signals ($\Delta F/F_0$) and SD signals of CCH-evoked responses from 12 cells. Peak fluorescence amplitudes were similar to mean values for 11 cells stimulated by strong photorelease of i-IP$_3$ ($\Delta F/F_0$ of 8.89 ± 0.3 for CCH vs. 7.27 ± 0.4 for i-IP$_3$); as were the rising phase kinetics of the global $Ca^{2+}$ signal (rise from 20% to 80% of peak 0.70 s ± 0.05 s for CCH vs. 0.80 s ± 0.06 s for i-IP$_3$). However, global $Ca^{2+}$ elevations evoked by CCH decayed more rapidly than those evoked by i-IP$_3$ (fall from 80% to 20% of peak 6.33 s ± 0.3 s for CCH vs 20.05 s ± 3.2 s for i-IP$_3$) - likely because the slowly-degraded i-IP$_3$ evoked a more sustained release of $Ca^{2+}$.

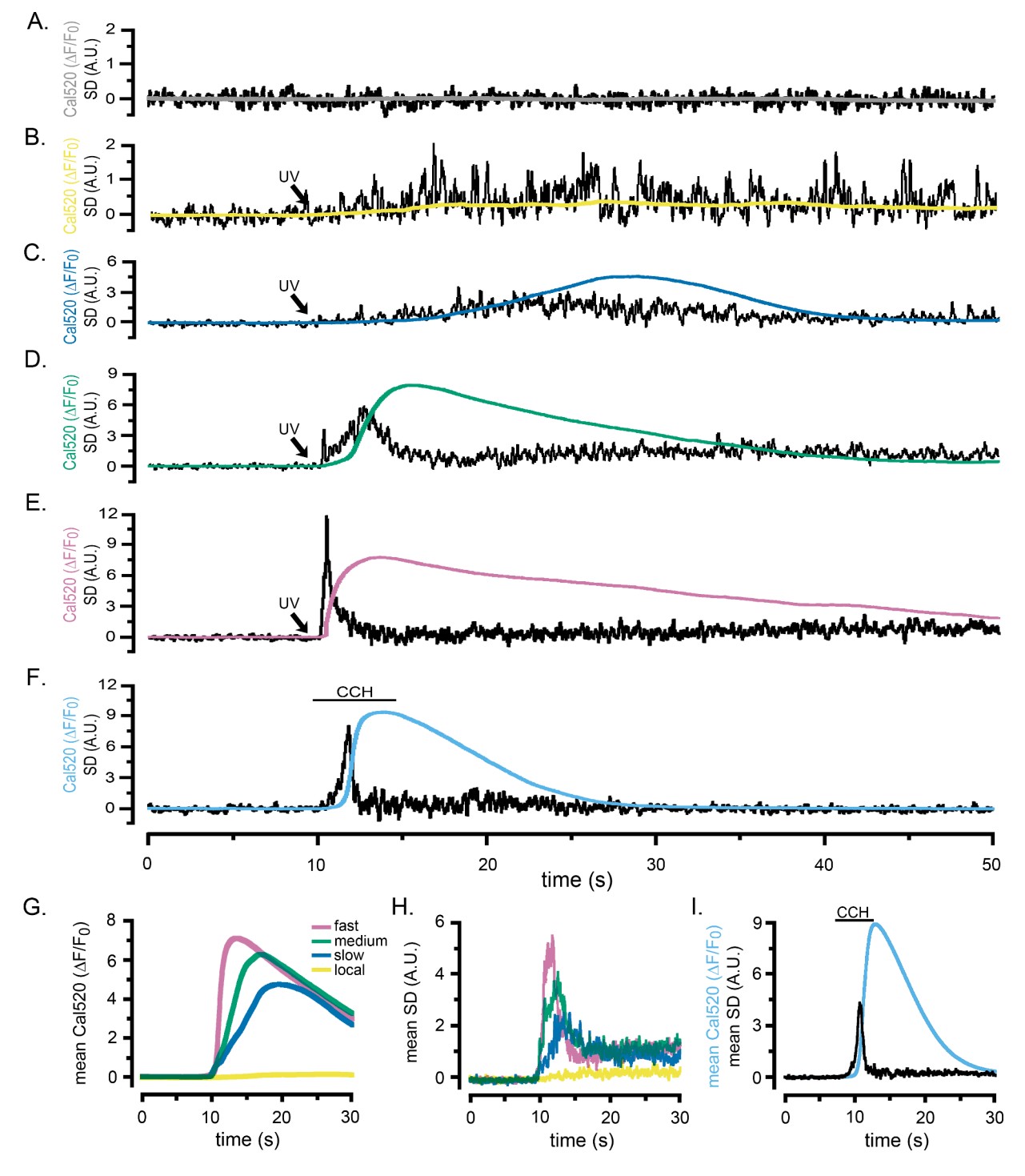

**Figure 2.** Localized fluctuations in cytosolic $[Ca^{2+}]$ occur predominantly during the rising phase of global $Ca^{2+}$ elevations. Representative records show the Cal520 fluorescence ratio ($\Delta F/F_0$; smooth traces) and the associated SD fluctuation measurements (noisy traces) from ROIs encompassing single WT HEK cells bathed in $Ca^{2+}$-free medium. (**A**) Record obtained under basal conditions without stimulation. (**B–E**) Responses evoked by progressively longer photolysis flashes to release increasing amounts of i-IP$_3$ in cells loaded with caged i-IP$_3$. The SD signals are presented in arbitrary units (A.U.) but are consistent throughout all panels. To better display responses to weaker stimuli, the y-axes are scaled differently between panels. (**F**) Responses evoked by application of carbachol (CCH; 10 µM) when indicated by the bar. (**G, H**) Pooled data plotting, respectively, means of the global $Ca^{2+}$ fluorescence signals and SD signals of cells stimulated with progressively increasing photorelease of i-IP$_3$ to evoke predominantly local $Ca^{2+}$ signals (yellow traces; n = 7), and global elevations with slow (blue; n = 9), medium (green; n = 13), and fast rising $Ca^{2+}$ signals (pink; n = 11). (**I**) Mean Cal520

*Figure 2 continued on next page*

Figure 2 continued

fluorescence ratio signal (cyan trace) and SD signal (black trace) averaged from 12 cells stimulated by local application of 10 µM CCH when marked by the bar.

The online version of this article includes the following video(s) for figure 2:

**Figure 2—video 1.** Detection of local $Ca^{2+}$ puffs by fluctuation analysis.
https://elifesciences.org/articles/55008#fig2video1
**Figure 2—video 2.** $Ca^{2+}$ fluctuations during global $Ca^{2+}$ elevations evoked by increasing photorelease of i-$IP_3$.
https://elifesciences.org/articles/55008#fig2video2
**Figure 2—video 3.** $Ca^{2+}$ fluctuations during a carbachol evoked global $Ca^{2+}$ signal.
https://elifesciences.org/articles/55008#fig2video3

## $Ca^{2+}$ puff activity terminates during the rising phase of global $Ca^{2+}$ signals

Puff activity (SD signal) showed a characteristic rise and fall during the rising phase of global $Ca^{2+}$ signals, and both parameters accelerated with increasing photorelease of i-$IP_3$ (*Figure 2*). To investigate the relationship between the bulk $Ca^{2+}$ level and puff activity in a time-independent manner, we took paired measurements of cell-wide SD signals and $Ca^{2+}$ level ($\Delta F/F_0$) at intervals during the rising phase of $IP_3$-evoked $Ca^{2+}$ elevations. *Figure 3A* shows a scatter plot of SD vs. $\Delta F/F_0$ values for measurements from the cell in *Figure 2D*, and *Figure 3B* plots corresponding mean data pooled from groups of cells that gave i-$IP_3$-evoked global signals with fast (pink circles), intermediate (green triangles) and slow (blue squares) rising phases. Although the amplitudes of the SD signals were greater for the faster rising responses, all cells showed similar 'inverted U' shaped relationships. In all three groups, the SD signal was maximal when the Cal520 fluorescence ratio reached a $\Delta F/F_0$ value of about two and then declined progressively as global $Ca^{2+}$ rose higher. This is illustrated more clearly in *Figure 3C*, where the curves for the three groups of cells superimpose closely after normalization to the same peak SD level. A closely similar inverted U relationship was observed for $Ca^{2+}$ elevations evoked by CCH (*Figure 3D*).

The decline in SD signal at higher $Ca^{2+}$ levels during global signals cannot be attributed to a failure of our algorithm to detect local fluctuations because of saturation of the Cal520 indicator dye. Notably, maximal fluorescence responses evoked by addition of ionomycin in high (10 mM) $Ca^{2+}$-containing medium ($\Delta F/F_0$ of $18.93 \pm 1.5$; n = 32 cells) considerably exceeded the peak fluorescence level evoked by even strong photorelease of i-$IP_3$ (mean $\Delta F/F_0$ $7.27 \pm 0.4$, n = 11 cells), and were greatly in excess of the fluorescence level ($\Delta F/F_0$ ~2; *Figure 3*) at which the SD signal began to decline. Moreover, we observed instances of local $Ca^{2+}$ signals even during large global $Ca^{2+}$ elevations ($\Delta F/F_0$ >8; *Figure 3—figure supplement 1*), and obtained SD signals using the lower affinity indicator fluo8L (Kd 1.86 µM vs. 320 nM for Cal520) confirming that puff activity was similarly suppressed prior to the peak of i-$IP_3$ evoked Ca (*Berridge et al., 2000*) elevations (*Figure 3—figure supplement 2*).

## $Ca^{2+}$ puffs are independent of extracellular $Ca^{2+}$

We performed the experiments in *Figures 1–3* using a bathing solution containing no added $Ca^{2+}$ together with 300 µM EGTA to specifically monitor the release of $Ca^{2+}$ from intracellular stores without possible confounding signals arising from entry of $Ca^{2+}$ across the plasma membrane. To explore whether these results were representative of responses in more physiological conditions, we examined $Ca^{2+}$ signals evoked by photoreleased i-$IP_3$ in WT HEK cells bathed in solutions containing 2 mM $Ca^{2+}$ (*Figure 3—figure supplement 3*). Cell-wide $Ca^{2+}$ responses and flurries of local $Ca^{2+}$ signals closely matched the patterns of activity in cells imaged in the absence of extracellular $Ca^{2+}$ (*Figure 3—figure supplement 3B–G*), and scatter plots of SD signal vs. global $Ca^{2+}$ fluorescence signal (*Figure 3—figure supplement 3H,I*) mirrored those in the absence of extracellular $Ca^{2+}$ (*Figure 3*). Thus, the puff activity during $IP_3$-evoked global $Ca^{2+}$ elevations appears independent of $Ca^{2+}$ influx into the cell. However, global $Ca^{2+}$ responses decayed more slowly when $Ca^{2+}$ was included in the bath solution (fall$_{80-20}$ for strong photoreleased i-$IP_3$ of 35.87 s $\pm$ 3.9 s in 2 mM $Ca^{2+}$ vs. 20.05 s $\pm$ 3.2 s in zero $Ca^{2+}$; Fall$_{80-20}$ for CCH of 13.06 s $\pm$ 0.5 s in 2 mM $Ca^{2+}$ vs 6.33 s $\pm$ 0.3 s in zero $Ca^{2+}$).

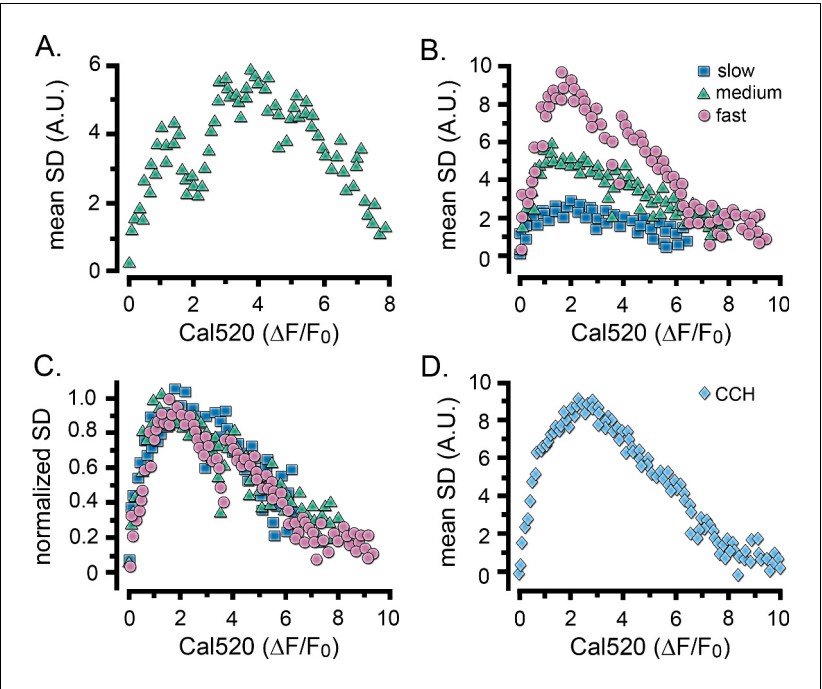

**Figure 3.** Relationship between $Ca^{2+}$ fluctuations and $Ca^{2+}$ level during the rise of global $Ca^{2+}$ signals. Scatter plots show measurements of the SD signal at intervals during the rising phase of global $Ca^{2+}$ response against the magnitude of the global $Ca^{2+}$ elevation ($\Delta F/F_0$) at that time. Data were binned at intervals of (0.1 $\Delta F/F_0$). (A) Measurements from the same cell as in *Figure 2D*. (B) Data from the same groups of cells as in *Figure 2G,H*, plotting mean SD signal amplitude as a function of mean $Ca^{2+}$ level during global responses for cells exhibiting slow (blue squares), intermediate (green triangles) and fast rising responses (pink circles). (C) The same data as in B, after normalizing to the respective maximum SD signals for each group of cells. (D) Scatter plot of mean SD signal amplitude as a function of $Ca^{2+}$ level during global responses for 12 cells stimulated by local application of CCH, as in *Figure 2I*.

The online version of this article includes the following figure supplement(s) for figure 3:

**Figure supplement 1.** Fluctuation image analysis resolves $Ca^{2+}$ puffs at peak elevations in global $Ca^{2+}$ signals.
**Figure supplement 2.** Temporal fluctuations in cytosolic [$Ca^{2+}$] during global $Ca^{2+}$ elevations in HEK cells monitored with the low affinity $Ca^{2+}$ indicator fluo8L.
**Figure supplement 3.** Temporal fluctuations in cytosolic [$Ca^{2+}$] during cell-wide $Ca^{2+}$ signals evoked by photoreleased of i-IP$_3$ in HEK cells bathed in medium including 2 mM $Ca^{2+}$.
**Figure supplement 4.** Patterns of $Ca^{2+}$ liberation during global $Ca^{2+}$ signals are substantially unaffected by depolarization of mitochondria and lysosomes with FCCP.

A likely explanation is that influx through slowly activating store-operated channels prolongs the response when extracellular $Ca^{2+}$ is present.

## Patterns of $Ca^{2+}$ release are largely unaffected by inhibition of mitochondrial and lysosomal $Ca^{2+}$ uptake

Mitochondria and lysosomes help shape intercellular $Ca^{2+}$ dynamics by accumulating and releasing $Ca^{2+}$ (*Rizzuto et al., 2012*; *Mammucari et al., 2018*; *Morgan et al., 2011*; *Yang et al., 2019*). To examine whether activity of these organelles influenced the spatial-temporal occurrence of puffs during IP$_3$-evoked global $Ca^{2+}$ signals, we treated WT HEK cells for 10 min with FCCP to inhibit mitochondrial (*Stout et al., 1998*; *Jensen and Rehder, 1991*) and lysosomal (*Churchill et al., 2002*) $Ca^{2+}$ uptake by dissipating the proton gradient necessary for $Ca^{2+}$ flux. (*Figure 3—figure supplement 4A,B*). Mean traces of whole-cell $Ca^{2+}$ fluorescence ($\Delta F/F_0$) and associated SD fluctuations in FCCP-treated cells stimulated with CCH exhibited local and global $Ca^{2+}$ signals similar to vehicle-treated controls, although with slightly smaller peak magnitudes (*Figure 3—figure supplement 4C,*

eLife Research article

*D*). Scatter plots of SD signal vs. bulk $Ca^{2+}$ level during the rising phase of CCH-evoked $Ca^{2+}$ elevations were closely similar in control and following FCCP application (*Figure 3—figure supplement 4E*).

## $Ca^{2+}$ puffs do not terminate because of rising cytosolic $Ca^{2+}$ during cell-wide elevations

In light of the resemblance between the inverted U relationship between puff activity and $Ca^{2+}$ level (*Figure 3*) and the well-known bell-shaped curve for biphasic modulation of $IP_3R$ channel activation by $Ca^{2+}$(*Iino, 1990*; *Bezprozvanny et al., 1991*), we considered whether the suppression of puff activity during global elevations might result because $IP_3Rs$ became inhibited by rising cytosolic $Ca^{2+}$ levels. To test this, we first examined the effect of elevating cytosolic $Ca^{2+}$ levels prior to evoking $IP_3$-mediated $Ca^{2+}$ signals. We loaded HEK WT cells with caged $Ca^{2+}$ (NP-EGTA) and delivered photolysis flashes of varying durations to cause jumps of cytosolic free $Ca^{2+}$ of different magnitudes before locally applying CCH from a puffer pipette (*Figure 4A*). Although the SD signals evoked by CCH declined progressively with increasing prior photorelease of $Ca^{2+}$, this reduction was matched by a similar diminution in peak amplitudes ($\Delta F/F_0$) of the global $Ca^{2+}$ signal. The open symbols in *Figure 4B* plot the ratio of puff activity (integral under SD traces) relative to the size of the CCH-evoked global $Ca^{2+}$ signal in each cell, and are presented after binning according to the magnitude of the preceding $Ca^{2+}$ jump evoked by photolysis of caged $Ca^{2+}$. Mean ratios (*Figure 4B*, filled symbols) remained almost constant for all $Ca^{2+}$ jumps; even at levels ($\Delta F/F_0 > 6$) corresponding to those where puff activity was strongly suppressed during the rising phase of global responses (*Figure 3*).

As a complementary approach, we then examined the effect of buffering the rise in cytosolic $[Ca^{2+}]$ during global responses by strong cytosolic loading of EGTA.

*Figure 4C* shows representative SD and $\Delta F/F_0$ traces in response to photoreleased i-$IP_3$ from a WT HEK cell that was loaded with EGTA by incubation for 1 hr with 15 μM EGTA-AM. The cell showed a typical flurry of puff activity like that in non-EGTA-loaded cells. Puffs ceased before the peak of the global $Ca^{2+}$ signal, even though the amplitude of the signal (2.5 $\Delta F/F_0$) was strongly attenuated. *Figure 4D* summarizes mean data from multiple cells, plotting paired measurements of cell-wide SD signals and $Ca^{2+}$ level ($\Delta F/F_0$) at intervals during the rising phase of $IP_3$-evoked $Ca^{2+}$ elevations, as in *Figure 3*. The data again followed an inverted U relationship (solid circles), but in comparison to control, non EGTA-loaded cells (open circles) the relationship was shifted markedly to the left. Notably, the peak SD signal was attained at a fluorescence level of about 0.40 $\Delta F/F_0$ vs. about 2 $\Delta F/F_0$ for controls, and puffs were substantially suppressed at fluorescence levels ($\Delta F/F_0 \sim 2$) where the puff activity was near maximal in control cells.

Taken together, these results demonstrate that inhibition of $IP_3Rs$ by elevated cytosolic $[Ca^{2+}]$ is not the primary mechanism causing puff activity to terminate during whole-cell $Ca^{2+}$ responses. They further buttress other evidence that the decline in SD signal during the rising phase of the response does not arise because the indicator dye becomes saturated, but faithfully reflects a physiological termination of puff activity.

## Partial depletion of ER $Ca^{2+}$ selectively inhibits $Ca^{2+}$ puffs

We next considered the possibility that puff activity may terminate during the rising phase of global $Ca^{2+}$ elevations because of falling luminal ER $[Ca^{2+}]$, rather than rising cytosolic $[Ca^{2+}]$. We tested this idea by imaging i-$IP_3$ evoked global $Ca^{2+}$ signals after partially depleting ER $Ca^{2+}$ stores while minimizing changes in cytosolic free $[Ca^{2+}]$.

In a first approach (*Figure 5A*), we transiently applied cyclopiazonic acid (CPA) to reversibly inhibit SERCA activity (*Uyama et al., 1992*), resulting in a net leak of $Ca^{2+}$ from the ER and a small elevation of cytosolic $Ca^{2+}$. Following wash-out of CPA, the cell was maintained in $Ca^{2+}$-free medium so that the cellular $Ca^{2+}$ content (including that of the ER) gradually depleted owing to passive and active extrusion across the plasma membrane. After about 4 min the resting cytosolic $Ca^{2+}$ level had returned close to the original baseline, and we delivered a photolysis flash to photorelease i-$IP_3$. This evoked a substantial elevation in global $Ca^{2+}$, yet the SD signal showed almost no transient puff activity during this response. Similar results were obtained in a further seven cells, as shown by the mean $\Delta F/F_0$ and SD traces in *Figure 5B*. To confirm that the suppression of puff activity resulted from cellular $Ca^{2+}$ depletion, we repeated this experiment, now making a paired comparison of

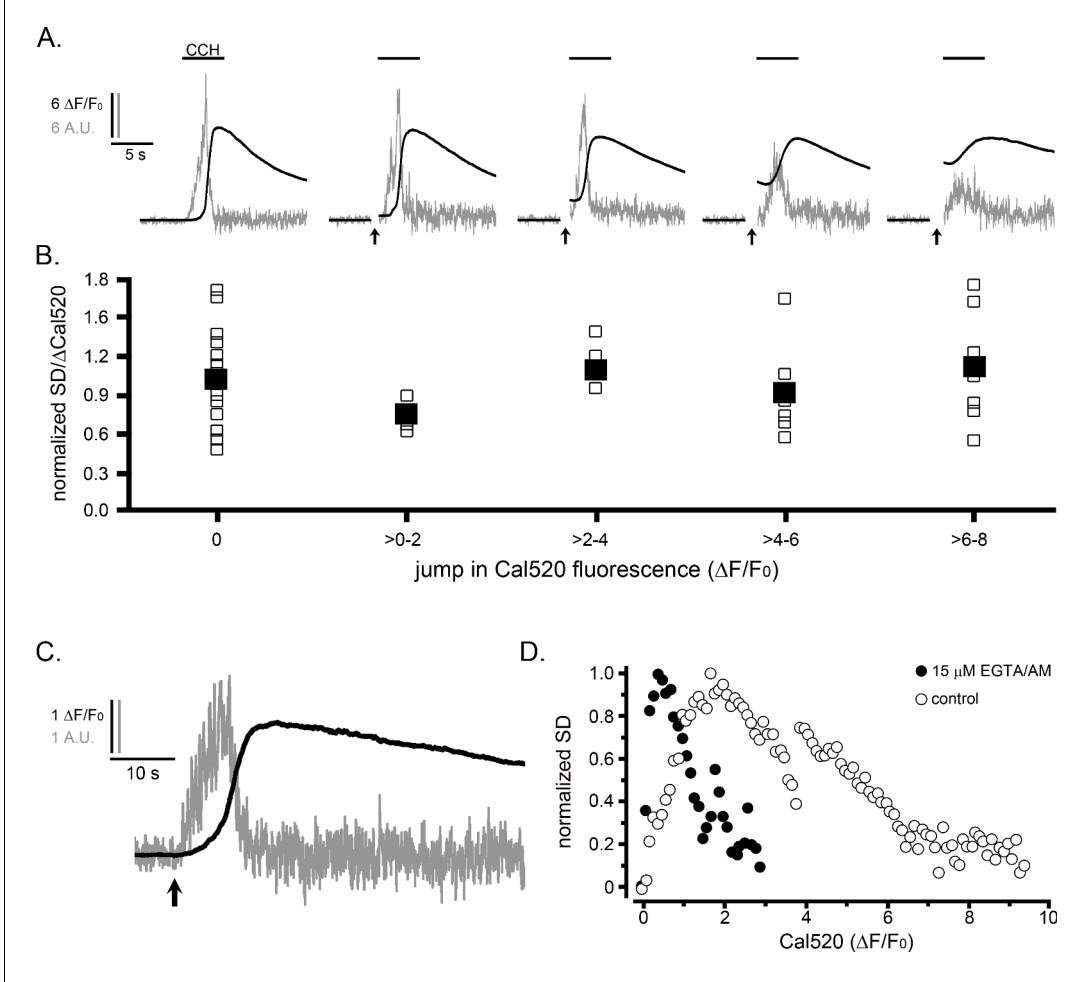

**Figure 4.** The suppression of $Ca^{2+}$ puffs during global signals does not result because of elevated cytosolic $[Ca^{2+}]$. (**A,B**) $IP_3$-evoked $Ca^{2+}$ puffs are not suppressed by prior photorelease of $Ca^{2+}$. (**A**) Traces depict fluorescence ratios (black; $\Delta F/F_0$) from WT HEK cells and corresponding SD signals (grey; in arbitrary units, A.U.). Records, from left to right, show responses from individual cells loaded with NP-EGTA (caged $Ca^{2+}$) that were unstimulated or exposed to increasing UV flash durations (marked by arrows) to photorelease progressively increasing amounts of free $Ca^{2+}$ before challenging cells with CCH (100 µM) locally delivered by a puffer pipette when indicated by the bars. Traces are blanked out during the artifact caused by the photolysis flash. (**B**) Data points from traces like those in A show the integral under SD trace (a measure of puff activity) as a ratio of the change in global $Ca^{2+}$ signal ($\Delta F/F_0$) evoked by CCH. The data are binned in terms of the jump in Cal520 fluorescence ($\Delta F/F_0$) evoked by photolysis of caged $Ca^{2+}$. Open symbols are from individual cells, and filled symbols are means for each group (respective n numbers for different bins; 20, 4, 4, 6, 8). Data are normalized with respect to the mean ratio without prior photorelease of $Ca^{2+}$. There was no significant difference between control CCH responses and CCH responses following $Ca^{2+}$ jumps (evaluated by Student T-test; p values between 0.17 and 0.66 for the different binned groupings). (**C,D**) Termination of puff activity is unaffected when global cytosolic $Ca^{2+}$ signals are attenuated by buffering with EGTA. (**C**) Traces showing the Cal520 fluorescence ratio ($\Delta F/F_0$; smooth trace) and SD signal (noisy trace) in response to photoreleased i-$IP_3$ in a representative WT HEK cell that was incubated with 15 µM EGTA/AM to buffer cytosolic $Ca^{2+}$ and attenuate the amplitude of the global $Ca^{2+}$ signal. (**D**) Scatter plots show measurements of the SD signal at intervals during the rising phase of global $Ca^{2+}$ responses against the magnitude of the global $Ca^{2+}$ elevation ($\Delta F/F_0$) at that time. Measurements were binned at intervals of (0.1 $\Delta F/F_0$) and SD data are normalized to a peak value of 1. Solid circles show mean data from 14 EGTA-loaded cells. For comparison, open circles present data reproduced from *Figure 3C* showing measurements from 11 control cells that gave fast rising responses to photoreleased i-$IP_3$.

i-$IP_3$-evoked responses between cells that were bathed for 30 min after washing out CPA either in $Ca^{2+}$-containing medium to allow ER store refilling (*Figure 5C*; *Figure 5—video 1*), or in $Ca^{2+}$-free medium (*Figure 5D*; *Figure 5—video 1*). Cells in both groups showed substantial global $Ca^{2+}$ responses that were not appreciably different in peak amplitudes (*Figure 5E*); but whereas the SD signals showed that puff activity was strongly suppressed in cells maintained in zero $Ca^{2+}$ medium,

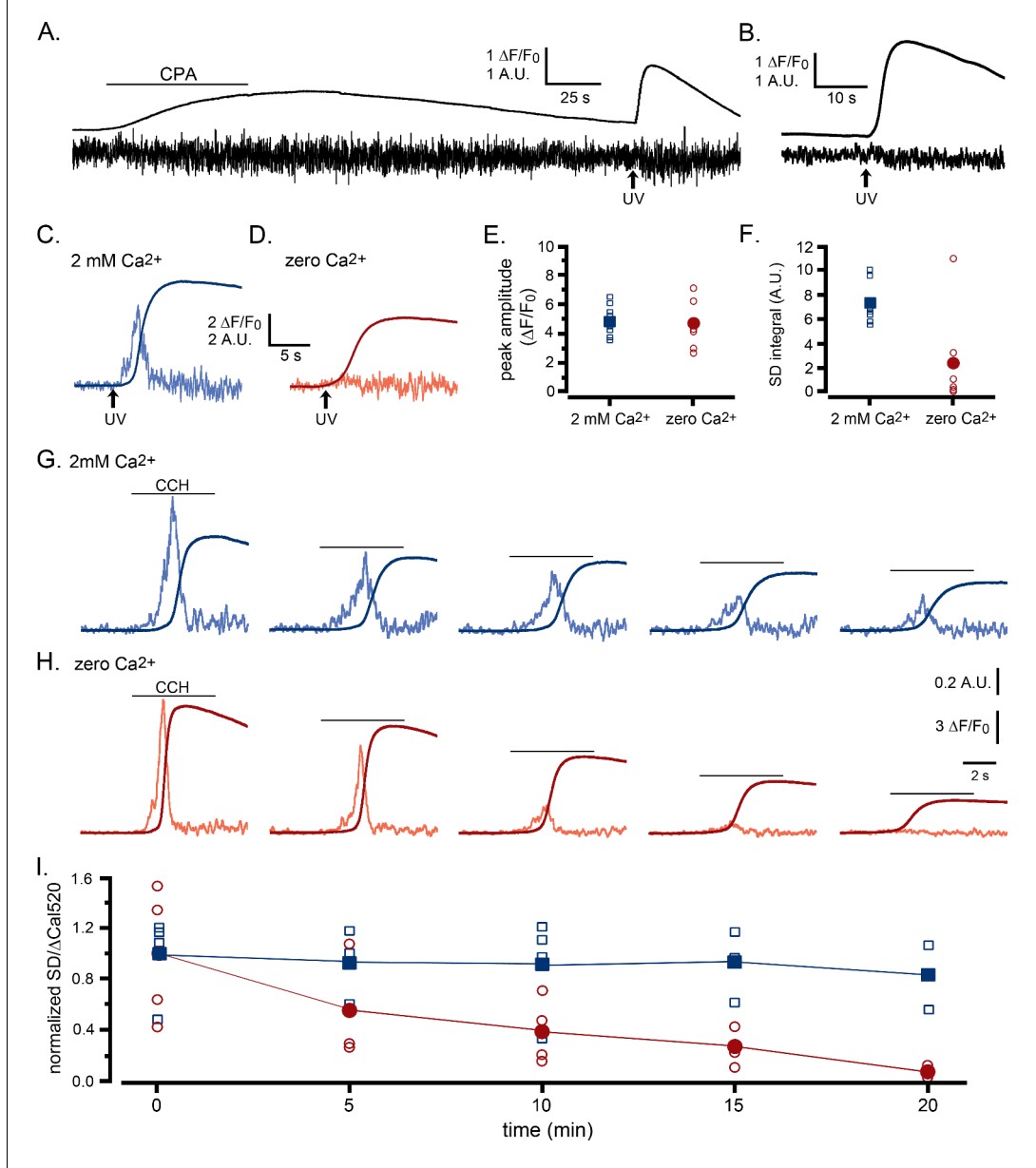

**Figure 5.** $Ca^{2+}$ puffs are selectively depressed by reduced ER $Ca^{2+}$ content. (**A–F**) Selective depression of puffs during i-IP$_3$-evoked global $Ca^{2+}$ signals following depletion of ER $Ca^{2+}$ content using transient application of cyclopiazonic acid (CPA; 50 µM) (**A**) The smooth trace shows fluorescence ratio ($\Delta F/F_0$) from a WT HEK cell, and the noisy trace the corresponding SD signal (in arbitrary units). The cell was bathed throughout in solution containing no added $Ca^{2+}$ and 300 µM EGTA, and CPA was locally applied from a puffer pipette during the time indicated by the bar. A UV flash was delivered when marked by the arrow to photorelease caged i-IP$_3$ loaded into the cell. (**B**) Mean $\Delta F/F_0$ and SD signals from seven WT HEK cells in response to photoreleased i-IP$_3$ following CPA treatment and wash in $Ca^{2+}$-free medium as in A. (**C,D**) Representative $\Delta F/F_0$ and SD responses to photoreleased i-IP$_3$ in individual cells that were bathed, respectively, in $Ca^{2+}$-containing or $Ca^{2+}$-free medium for 30 min following treatment with CPA as in A. (**E**) Peak amplitudes of global fluorescence signals evoked by photoreleased i-IP$_3$ in experiments like those in C,D, for cells bathed in $Ca^{2+}$-containing (n = 8 cells; blue squares) or $Ca^{2+}$-free medium (n = 6; red circles). Open symbols denote measurements from individual cells; filled symbols are means. No significant difference between peak amplitudes ($\Delta F/F_0$) of cells bathed in $Ca^{2+}$-containing and $Ca^{2+}$-free medium (Student T test; p=0.72). (**F**) Corresponding measurements of integral under SD traces (puff activity) during the time from the photolysis flash to the peak global fluorescence signal. SD integrals were significantly different between cells bathed in $Ca^{2+}$-containing and $Ca^{2+}$-free medium (Student T test; p=0.012). (**G–I**) Selective depression of puffs by depleting ER $Ca^{2+}$ content by repeated applications of CCH in zero $Ca^{2+}$ bathing solution. (**G,H**) Global $Ca^{2+}$ signals (smooth traces; $\Delta F/F_0$) and SD signals (noisy traces) evoked by successive, identical applications of CCH at 5 min intervals in two representative cells bathed, respectively, in medium containing 2 mM $Ca^{2+}$ or 300 µM EGTA with no added $Ca^{2+}$. Amplitudes of the SD signals are depicted after normalizing to the peak amplitude of the first response for each cell. (**I**) Data points show the ratio of puff activity (integral under the SD trace) vs. peak magnitude of

*Figure 5 continued on next page*

Figure 5 continued

the global $Ca^{2+}$ signal ($\Delta F/F_0$) for successive responses evoked by CCH application at 5 min intervals. Blue squares are data from cells bathed in medium containing 2 mM $Ca^{2+}$ and red circles are from cells in $Ca^{2+}$-free medium; open symbols are ratios from individual cells and filled symbols are means. Data are plotted after normalizing to the mean SD integral and peak $\Delta F/F_0$ evoked by the initial stimulus in each condition. Responses were significantly different between cells bathed in the presence and absence of external $Ca^{2+}$ for times $\geq$ 10 min (Student T test; p=0.000008).

The online version of this article includes the following video for figure 5:

**Figure 5—video 1.** Partial depletion of ER $Ca^{2+}$ selectively inhibits $Ca^{2+}$ puff activity.

https://elifesciences.org/articles/55008#fig5video1

cells in $Ca^{2+}$-containing medium showed robust puff activity during the rising phase of the response (*Figure 5F*).

As an alternative approach to partially deplete ER $Ca^{2+}$ without pharmacological intervention, we evoked $Ca^{2+}$ signals by repeated applications of CCH at 5 min intervals, and compared responses in cells bathed in $Ca^{2+}$-containing (*Figure 5G*) and $Ca^{2+}$-free solutions (*Figure 5H*). In both cases, the amplitudes of the global $Ca^{2+}$ signals progressively declined, likely a result of inhibition of IP$_3$Rs. However, whereas the amplitude of puff activity reported by SD signals in cells bathed in $Ca^{2+}$-containing medium fell roughly in proportion to the amplitude of the global fluorescence signal, puff activity in $Ca^{2+}$-free medium declined abruptly. In the example depicted in *Figure 5H*, no activity was evident in the SD signal after the fifth stimulus at 20 min even though an appreciable global $Ca^{2+}$ elevation remained. To quantify these data, we determined puff activity as the integral under the SD trace, and plotted the normalized ratio of puff activity vs. peak global $Ca^{2+}$ amplitude (*Figure 5I*). For cells in $Ca^{2+}$-containing medium, the mean ratio remained constant across successive stimuli (blue squares, *Figure 5I*), whereas it declined almost to zero for cells in $Ca^{2+}$-free medium (red circles, *Figure 5I*).

We conclude from these results that $Ca^{2+}$ puff activity is modulated by ER $Ca^{2+}$ store content, and that when stores are partially depleted IP$_3$ can still evoke $Ca^{2+}$ release by a process that is independent of puff activity, and occurs without detectable temporal fluctuations. We term this mode of $Ca^{2+}$ liberation as 'diffuse' release and refer to $Ca^{2+}$ puffs as a 'punctate' mode of $Ca^{2+}$ liberation.

## All three IP$_3$R isoforms mediate punctate and diffuse modes of $Ca^{2+}$ liberation

In common with many other cell types, WT HEK and HeLa cells express all three major IP$_3$R isoforms – types 1, 2, and 3 – that are encoded by separate genes and translated into structurally and functionally distinct proteins that co-translationally oligomerize to form heterotetrameric channels. We (*Lock et al., 2018*) and others (*Mataragka and Taylor, 2018*) recently demonstrated that all three isoforms can individually mediate $Ca^{2+}$ puffs. We now utilized HEK cells genetically engineered to express single IP$_3$R isoforms to evaluate the respective roles of each isoform in liberating $Ca^{2+}$ via punctate, localized transients versus sustained, diffuse release.

We evoked $Ca^{2+}$ liberation in WT HEK cells and cells exclusively expressing type 1, 2, or 3 IP$_3$Rs by local application of CCH (*Figure 6*). All three single-isoform-expressing cell lines exhibited patterns of responses qualitatively similar to WT cells. The SD traces showed flurries of puffs during the foot and rising phase of global $Ca^{2+}$ signals that ceased before the time of the peak global $Ca^{2+}$ elevation (*Figure 6A–H*). Nevertheless, notable differences were apparent between the isoforms. Cells expressing IP$_3$R1 generated whole-cell $Ca^{2+}$ signals having much smaller amplitudes and slower rising phases than WT and R2- and R3-expressing cells, and localized fluctuations persisted longer (*Figure 6C,D,I*). In contrast, IP$_3$R2-expressing cells displayed fast rising, large amplitude $Ca^{2+}$ signals, with a transient flurry of $Ca^{2+}$ fluctuations concentrated during the initial portion of the rising phase (*Figure 6E,F*). $Ca^{2+}$ signals in cells expressing IP$_3$R3 (*Figure 6G,H*) were similar in amplitude to WT and IP$_3$R2-expressing cells, but with slower rates of rise and more prolonged flurries of puffs. Scatter plots of puff activity (SD signal) as a function of the global $Ca^{2+}$ level ($\Delta F/F_0$) during the rising phase of the global response further highlighted these differences (*Figure 6J,K*). $Ca^{2+}$ fluctuations were maximal when Cal520 fluorescence ($\Delta F/F_0$) rose to roughly 1.5, 6, and 3 for types 1, 2, and 3 IP$_3$Rs, respectively; and similarly large differences were evident in the global $Ca^{2+}$ level attained when puff activity terminated.

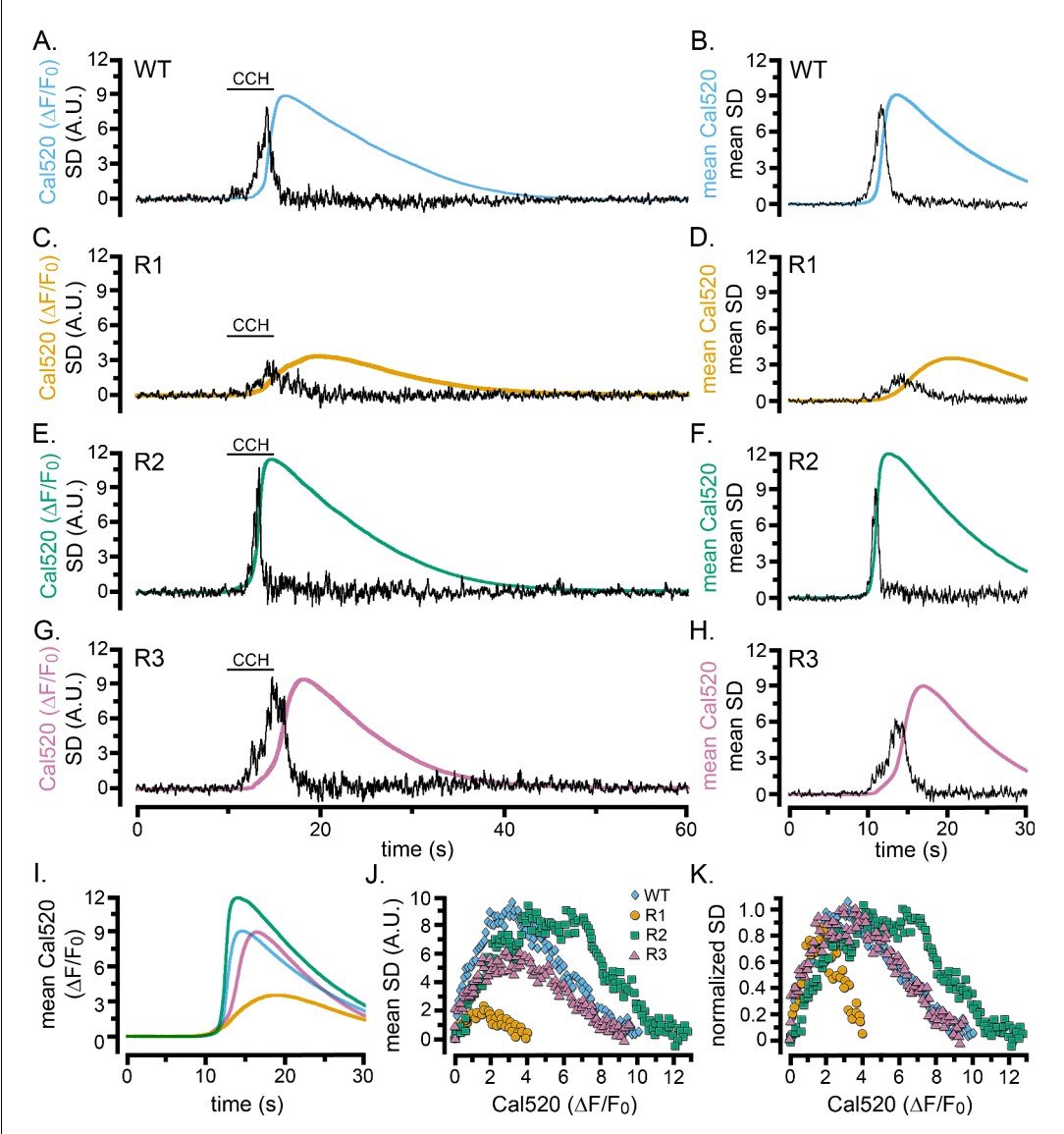

**Figure 6.** Cell-wide $Ca^{2+}$ elevations and SD fluctuations in WT HEK cells and cells exclusively expressing single IP₃R isoforms. (A–H) Traces show whole-cell Cal520 fluorescence ratio (smooth colored traces; $\Delta F/F_0$) and SD fluctuations (noisy black traces) of HEK cells locally stimulated with CCH locally delivered in a $Ca^{2+}$-containing bath solution when indicated by the bars. Panels on the left are representative records from individual cells, and panels on the right show mean traces from 7 (B) or 3 (D,F,H) cells. (A, B) Records from HEK WT cells. (C–H) Records from HEK cells solely expressing IP₃R1 (C, D), IP₃R2 (E,F), or IP₃R3 (G,H). (I) Overlaid mean Cal520 fluorescence ratio traces, aligned to their rising phase, in WT HEK cells (cyan; n = 7), and HEK cells solely expressing IP₃R1 (gold; n = 3), IP₃R2 (green; n = 3), and IP₃R3 (pink; n = 3). (J) Scatter plots of SD signal vs. fluorescence ratio during the rising phase of the $Ca^{2+}$ responses in WT HEK cells (cyan diamonds) and HEK cells solely expressing IP₃R1 (gold circles) IP₃R2 (green squares) or IP₃R3 (pink triangles). Data points are means from the same cells as in I. (K) The same data as in J, after normalizing to the same peak SD values.
The online version of this article includes the following figure supplement(s) for figure 6:

**Figure supplement 1.** $Ca^{2+}$ fluctuations during global $Ca^{2+}$ signals in HeLa cells.

## HeLa and HEK cells exhibit similar patterns of $Ca^{2+}$ signals

We utilized HEK cells for most experiments because of the availability of cell lines expressing individual IP₃R isoforms (*Alzayady et al., 2016*). The patterning of local, transient $Ca^{2+}$ signals during IP₃-mediated whole-cell $Ca^{2+}$ elevations was not unique to this cell type. Stimulation of HeLa cells with histamine also evoked global $Ca^{2+}$ signals accompanied by flurries of local $Ca^{2+}$ activity during the rising phase, which subsided as $Ca^{2+}$ levels continued to rise (*Figure 6—figure supplement 1*).

# Diffuse Ca$^{2+}$ signals in TIRF do not reflect punctate release in the cell interior

The data in *Figures 1–6* derive from TIRF imaging of Ca$^{2+}$ signals in close proximity to the plasma membrane, where a majority (~80%) of puff sites in WT HEK cells are located (*Lock et al., 2018*). However, TIRF microscopy provides no direct information from the interior of the cell, leaving open the question as to whether slow diffusion of Ca$^{2+}$ ions from puffs at internal sites may contribute to the diffuse component of the Ca$^{2+}$ signal visualized in TIRF images after the puff flurry has ceased. To address this issue, we applied fluctuation analysis to images obtained using lattice light-sheet (LLS) microscopy to record Ca$^{2+}$ signals within diagonal optical 'slices' through the cell volume (*Ellefsen and Parker, 2018*).

*Figure 7A,B* illustrate LLS Ca$^{2+}$ fluorescence ratio images and corresponding SD images recorded before and after photorelease of i-IP$_3$ to evoke a global Ca$^{2+}$ response. Similar to observations with TIRF imaging, the SD images revealed local Ca$^{2+}$ transients that began soon after photorelease, and before any appreciable rise in the global Ca$^{2+}$ signal (*Figure 7A*, panel ii). Discrete events then continued during much of the rising phase of the global signal (panels iii-v) but had largely ceased at the time of the peak global signal (panel vi). In this cell Ca$^{2+}$ puffs were primarily restricted to the cell periphery, whereas *Figure 7B* shows an example from another cell where local activity was observed both around the periphery and in the cell interior.

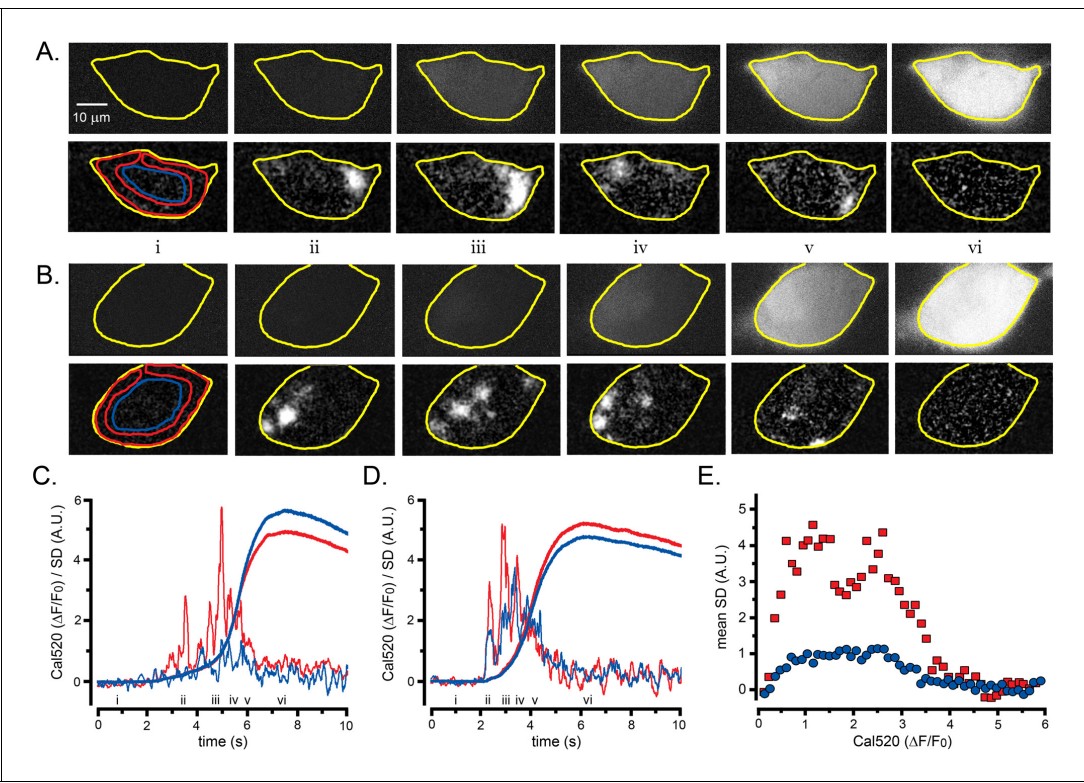

**Figure 7.** Lightsheet imaging of global Ca$^{2+}$ elevations evoked in HEK cells by photoreleased i-IP$_3$. (**A**) Upper panels show 45° diagonal image 'slices' through the center of a WT HEK cell imaged by lattice light-sheet microscopy. Grey scale intensities correspond to increases in fluorescence (ΔF) of Cal520 relative to the mean intensity (F$_0$) averaged over 100 frames before stimulation (ΔF/F$_0$). Each panel is a single 10 ms exposure, captured at times before and after stimulation, as indicated by the Roman numerals in C. The cell outline is marked in yellow. Lower panels show corresponding SD images, at times corresponding to the upper panels. Colored outlines mark ROIs used to derive ΔF/F$_0$ and SD traces from peripheral (red) and center (blue) regions of the cell. (**B**) Corresponding ΔF/F$_0$ and SD lightsheet images from a different HEK cell that showed more prominent puff activity in the center of the cell. (**C**) Measurements of ΔF/F$_0$ (smooth traces) and SD (noisy traces) from the cell illustrated in A. Traces in red show average measurements from the peripheral region of interest marked in the bottom left panel of A, and traces in blue show measurements from the central region of interest. (**D**) Corresponding measurements of ΔF/F$_0$ and SD from the cell illustrated in B. (**E**) Scatter plot of SD signal versus Ca$^{2+}$ fluorescence (ΔF/F$_0$) at intervals during the rising phase of global Ca$^{2+}$ signals. Data are from eight cells, with measurements binned at intervals of 0.1 ΔF/F$_0$.

*Figure 7C,D* shows respective measurements from these two cells, plotting fluorescence ratio changes ($\Delta F/F_0$) and SD signals from ROIs encompassing peripheral (red traces) and central (blue traces) regions of the cells, as indicated in the leftmost lower panels of *Figure 7A,B*. In both cells, the local $Ca^{2+}$ activity monitored by SD fluctuations started within a few hundred ms of the photolysis flash and was maximal during the early portion of the rising phase. The SD signal then declined, returning close to baseline as the global $Ca^{2+}$ signal approached a peak. For the cell illustrated in *Figure 7A*, the SD signal within the peripheral region was much greater than in the central region, even though the rise in global $Ca^{2+}$ was slightly smaller. In contrast, the cell illustrated in *Figure 7B* showed a SD signal in the interior that was similar in size to the periphery (*Figure 7D*). On average, however, mean SD signals from the cell interior were about one quarter of that at the periphery, and fluctuations arising from interior sites followed a similar relation with bulk $Ca^{2+}$ level as peripheral sites (*Figure 7E*).

Given the relatively low average level of puff activity in the cell interior, and the similar termination of internal and peripheral puff flurries during the rising of global $Ca^{2+}$ signals, we conclude that the diffuse component of the $Ca^{2+}$ rise observed by TIRF microscopy cannot be accounted for by $Ca^{2+}$ spreading from punctate release at internal sites and becoming blurred by diffusion in space and time.

## Puff activity contributes only a fraction of the total $Ca^{2+}$ liberated during global signals

To assess the relative contributions of punctate versus diffuse modes of $Ca^{2+}$ release during global $Ca^{2+}$ signals, we derived the kinetics of $Ca^{2+}$ flux into the cytosol through $IP_3Rs$ on the basis that the cell-wide fluorescence signal reflects a balance between $Ca^{2+}$ release into the cytoplasm and its subsequent removal. To obtain a rate constant for removal of cytosolic $Ca^{2+}$ in WT HEK cells, we recorded the decline of fluorescence $Ca^{2+}$ signals following transient photorelease of $Ca^{2+}$ from caged $Ca^{2+}$ loaded into the cytosol (*Figure 8—figure supplement 1A*), and during the final 'tail' of CCH-evoked $Ca^{2+}$ signals when $Ca^{2+}$ liberation would have almost ceased (*Figure 8—figure supplement 1B*). Both fitted well to single exponential decay functions, consistent with a dominantly first order removal process, with respective mean rate constants of 0.22 and 0.32 $s^{-1}$.

We then calculated the instantaneous $Ca^{2+}$ release flux at intervals throughout the time course of a global $Ca^{2+}$ response by differentiating the whole-cell fluorescence $Ca^{2+}$ signal and adding to this the estimated rate of $Ca^{2+}$ removal; for i-$IP_3$ signals we used a rate constant of 0.22 $s^{-1}$; for CCH evoked responses we applied rate constants (0.3 $s^{-1}$ to 0.6 $s^{-1}$) that were determined from the tail-end of the global $Ca^{2+}$ decay for that particular cell.

We used data from the experiment of *Figure 5C,D* to compare the kinetics of $Ca^{2+}$ liberation during $Ca^{2+}$ signals under normal conditions, and when puff activity had been inhibited by partial depletion of ER $Ca^{2+}$ store content. *Figure 8A,B* show records from two representative cells that gave global $Ca^{2+}$ responses of comparable peak amplitudes (black traces). However, whereas the SD signals (grey traces) exhibited the normal flurry of puff activity in the control cell (*Figure 8A*) this activity was almost completely suppressed in the cell pretreated with CPA (*Figure 8B*). The red traces show the respective rates of $Ca^{2+}$ release into the cytosol, revealing a larger initial transient of $Ca^{2+}$ liberation in the control cell during the flurry of puff activity. *Figure 8C* shows overlaid mean traces of $Ca^{2+}$ release from control (n = 5) and CPA-treated cells (n = 6). Colored areas indicate the relative cumulative amounts of $Ca^{2+}$ entering the cytosol (integral under the release trace) in CPA-treated cells where puff activity was substantially abolished (blue shading), and the additional $Ca^{2+}$ flux (pink shading) in control cells showing flurries of puffs. From these respective areas, we estimate that, in normal conditions, the punctate liberation of $Ca^{2+}$ through puff activity contributes about 41% of the total $Ca^{2+}$ release responsible for the initial rise of $Ca^{2+}$ toward its peak. *Figure 8D,E* further illustrate representative records of SD signals (grey traces), global $Ca^{2+}$ (black), rate of $Ca^{2+}$ release into the cytosol (red), and cumulative amount of $Ca^{2+}$ released (blue) during the entire time course of global $Ca^{2+}$ signals evoked by photoreleased i-$IP_3$ (*Figure 8D*) and by CCH (*Figure 8E*). Because much of the cumulative $Ca^{2+}$ release through $IP_3Rs$ arises from a sustained, low level flux that continues after the peak, $Ca^{2+}$ puffs on average contribute only about 13% of the total $Ca^{2+}$ liberation during global i-$IP_3$-evoked signals, and about 17% during shorter-lasting responses evoked by CCH (*Figure 8F*).

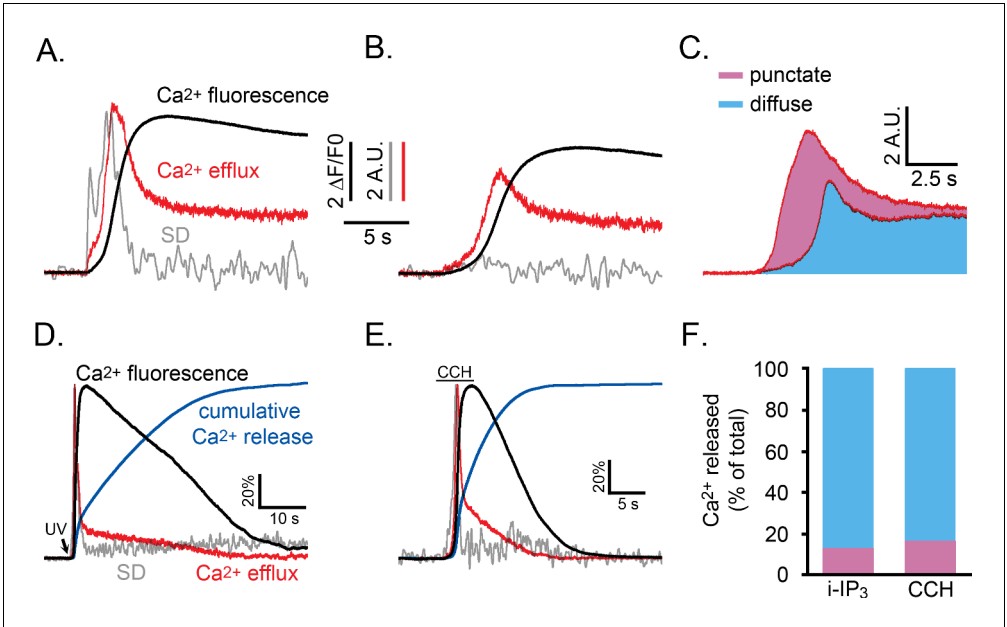

**Figure 8.** Relative proportions of $Ca^{2+}$ released by punctate versus diffuse modes of $Ca^{2+}$ liberation during an $IP_3$-evoked global $Ca^{2+}$ signal. (**A,B**) Whole cell $Ca^{2+}$ fluorescence responses (black traces) and associated SD signals (grey traces) during the initial phase of a $Ca^{2+}$ response evoked by photoreleased i-$IP_3$ in representative WT HEK cells. The red traces show the estimated rate of $Ca^{2+}$ efflux, derived as described in the text. Both panels show responses from cells pretreated with CPA as in *Figure 5A* that were treated identically, except that the cell in A was incubated in $Ca^{2+}$-containing medium to allow refilling of ER $Ca^{2+}$, whereas the cell in B was incubated in $Ca^{2+}$-free medium to maintain the ER $Ca^{2+}$ in a partially depleted state and suppress puff activity. The SD and $Ca^{2+}$ efflux traces in A are scaled to similar peak height for clarity; traces in B are scaled the same as in A. (**C**) Mean $Ca^{2+}$ efflux traces from five cells in $Ca^{2+}$-containing medium that showed robust puff activity (top) and six cells in $Ca^{2+}$-free medium where puff activity was almost absent (lower). The area shaded blue reflects the relative amount of $Ca^{2+}$ released when puff activity was absent, and the pink area reflects the additional amount of $Ca^{2+}$ release attributable to $Ca^{2+}$ puffs. (**D,E**) Traces show SD signal (grey), global $Ca^{2+}$ fluorescence ratio (black) and calculated $Ca^{2+}$ efflux rate (red) for the entire duration of responses evoked by photoreleased i-$IP_3$ (**D**) and by CCH (**E**). Blue traces additionally show the cumulative percentage of $Ca^{2+}$ released, derived by integrating under the red $Ca^{2+}$ efflux traces. For clarity of presentation all traces are shown scaled to the same peak height. (**F**) Bars show mean percentages of total $Ca^{2+}$ release during i-$IP_3$-evoked (left; n = 8 cells) and CCH-evoked signals (right; n = 7 cells) under control conditions attributable to punctate (pink) and diffuse (blue) modes of $Ca^{2+}$ liberation. Data were calculated from the cumulative $Ca^{2+}$ release at the time puff activity had ceased in each cell, assuming 41% of that release was due to punctate release.

The online version of this article includes the following figure supplement(s) for figure 8:

**Figure supplement 1.** Protocols to estimate the rate of $Ca^{2+}$ clearance from the cytosol.

## Discussion

$Ca^{2+}$ puffs are transient, localized elevations in cytosolic $Ca^{2+}$ that arise from concerted opening of small numbers of $IP_3$Rs clustered at fixed intracellular sites (*Parker and Yao, 1991*; *Thillaiappan et al., 2017*). Puffs are apparent as discrete events superimposed on a steady basal $Ca^{2+}$ level when cytosolic $IP_3$ concentrations are modestly elevated (*Parker and Yao, 1991*; *Yao et al., 1995*; *Parker et al., 1996*), whereas higher concentrations of $IP_3$ evoke global, cell-wide $Ca^{2+}$ signals on which puffs are evident on the rising phase (*Bootman et al., 1997a*; *Marchant et al., 1999*). Puffs have been proposed as fundamental building blocks of $IP_3$-mediated $Ca^{2+}$ signaling (*Bootman et al., 1997a*; *Parker et al., 1996*; *Berridge, 1997*; *Bootman and Berridge, 1996*; *Marchant, 2001*; *Marchant et al., 1999*; *Mataragka and Taylor, 2018*); acting as local signals in their own right at low [$IP_3$] and mediating global $Ca^{2+}$ signals at higher [$IP_3$] by a fire-diffuse-fire mechanism whereby $Ca^{2+}$ released by a puff site diffuses to activate CICR at neighboring sites (*Bootman et al., 1997a*; *Parker et al., 1996*; *Berridge, 1997*; *Dawson et al., 1999*). However,

it has been difficult to definitively test this 'building block' model because puffs become obscured by the large global $Ca^{2+}$ elevations; and recent theoretical simulations have questioned whether the summation of $Ca^{2+}$ released through coordinated puff activity at multiple sites is alone sufficient to propagate global cytosolic $Ca^{2+}$ signals (*Piegari et al., 2019*). Here, we addressed this topic by analyzing temporal and spatial fluctuations in $Ca^{2+}$ image data to resolve local $Ca^{2+}$ transients during global signals (*Swaminathan et al., 2020*). Our main conclusion is that global $Ca^{2+}$ signals involve two modes of $Ca^{2+}$ liberation through $IP_3Rs$: 'punctate' release as a flurry of transient, local events, and a more sustained, 'diffuse' release mode.

As with any new approach, we first needed to validate the ability of our algorithm to faithfully report local $Ca^{2+}$ transients during even large global $Ca^{2+}$ elevations, when resolution may be impaired by factors including the dynamic range of the indicator dye and by increased photon shot noise at high fluorescence levels. A particular concern was whether the indicator (Cal520) we used for most experiments may have approached saturation, thus 'clipping' the signals to artifactually suppress the temporal SD signal and giving a false impression that puff activity terminates as the $Ca^{2+}$ level and fluorescence rise during global signals. Several lines of evidence convincingly argue that this is not the case. Notably: (i) maximal, saturating signals evoked by ionomycin (~19 $\Delta F/F_0$) were much higher than mean peak $IP_3$-evoked fluorescence signals (~7 $\Delta F/F_0$), and puff activity began to decline as fluorescence rose above ~2 $\Delta F/F_0$; (ii) we observed patterns of puff activity using the low affinity indicator fluo-8L ($K_d$1.86 μM) that closely matched those obtained with Cal520 ($K_d$320 nM) (*Figure 3—figure supplement 2*); (iii) we were able to resolve instances of local puff activity even at the peak of $IP_3$-evoked global fluorescence elevations (*Figure 3—figure supplement 1*); (iv) the kinetics of puff activity were closely similar in cell lines individually expressing single $IP_3R$ isoforms, despite large differences in the amplitudes of the global $Ca^{2+}$ signals (*Figure 6*); (v) the onset and termination of puff activity during the rise of $IP_3$-mediated global $Ca^{2+}$ signals were little altered when the initial basal $Ca^{2+}$ level was elevated (*Figure 4A,B*) or, conversely, when the global $Ca^{2+}$ rise was attenuated by buffering with cytosolic EGTA (*Figure 4C,D*). Finally, the suppression of punctate $Ca^{2+}$ liberation throughout all phases of $IP_3$-evoked $Ca^{2+}$ responses when ER $Ca^{2+}$ stores were partially depleted (*Figure 5*) strongly supports our proposal that $Ca^{2+}$ liberation can arise in a diffuse manner, independent of local puff events.

## Puff activity during global $Ca^{2+}$ signals

In agreement with previous findings (*Bootman et al., 1997a*; *Marchant et al., 1999*) our fluctuation analyses reveal flurries of puffs during the initial rise of $IP_3$-mediated global $Ca^{2+}$ signals. However, although puff activity was evident during the initial foot of the response and peaked early during the rising phase, it then subsided during the later portion of the rising phase, with few or no transient, local $Ca^{2+}$ signals evident by the time of the peak. Notably, overall $Ca^{2+}$ levels continued to rise after puffs had largely ceased, and cytosolic $Ca^{2+}$ remained elevated for several seconds in the face of rapid removal from the cytosol, during which time $Ca^{2+}$ fluctuations were largely suppressed.

This 'noise-free' component of the fluorescence signal cannot be attributed to slow diffusion of $Ca^{2+}$ liberated as puffs to fill in spaces between release sites. Diffusion would be rapid (e.g. mean time of ~300 ms to diffuse 5 μm assuming an effective diffusion coefficient of 20 μm² s⁻¹). In any case, the average fluorescence signal would not be expected to increase appreciably if the total amount of $Ca^{2+}$ in the imaging volume remained constant. Utilizing lightsheet imaging we further excluded the possibility that the continuing rise in near-plasmalemmal $Ca^{2+}$ observed by TIRF imaging might arise through diffusion of $Ca^{2+}$ over longer distances after liberation at sites in the cell interior. Finally, as noted previously, the observation of large global $Ca^{2+}$ signals in the absence of detectable fluorescence fluctuations (*Figure 5*) definitively points to a mode of $Ca^{2+}$ liberation that is independent of puff activity.

By deriving the time course of cumulative $Ca^{2+}$ liberation during global responses we estimated that puffs contribute only ~41% of the initial $Ca^{2+}$ flux that drives the peak of the $Ca^{2+}$ response, and an even smaller proportion (~15%) of the cumulative flux during its entire time course. Thus, a second component of continuous, spatially diffuse release of intracellular $Ca^{2+}$ is responsible for generating and sustaining a large part of whole-cell $Ca^{2+}$ signals. These two components are further discriminated by procedures that selectively promoted either puff activity (*Dargan and Parker, 2003*; *Smith et al., 2009*) (by cytosolic loading of slow $Ca^{2+}$ buffers), or global $Ca^{2+}$ elevations in the absence of localized $Ca^{2+}$ transients (by partial depletion of ER $Ca^{2+}$ store content). We term these

two modes of IP$_3$-mediated Ca$^{2+}$ release as 'punctate' (discontinuous in time and space) and 'diffuse' (smoothly varying in time and space).

Small elevations of [Ca$^{2+}$] promote opening of IP$_3$R channels (*Iino, 1990*; *Bezprozvanny et al., 1991*) and increase puff frequency (*Yamasaki-Mann et al., 2013*). The accelerated puff activity during the foot and initial upstroke of the global Ca$^{2+}$ signal is thus likely a consequence of a rising basal cytosolic Ca$^{2+}$ level, resulting both from the puffs themselves and from diffuse Ca$^{2+}$ liberation. This positive feedback of cytosolic Ca$^{2+}$ to promote opening of IP$_3$Rs is inherently regenerative, so it is imperative that mechanisms exist to 'put out the fire'. Our results illuminate at least two mechanisms, acting across different time scales, to terminate punctate Ca$^{2+}$ liberation through IP$_3$Rs. Individual puffs terminate rapidly as IP$_3$R channels close within tens of ms (*Parker et al., 1996*; *Bootman et al., 1997b*) via stochastic inhibition by high local Ca$^{2+}$ levels (*Shuai et al., 2008*) and potential allosteric interactions between clustered IP$_3$Rs at a puff site (*Wiltgen et al., 2014*). However, during the rising phase of a global Ca$^{2+}$ signal, each IP$_3$R cluster may generate a flurry of repeated puffs - indicating that although the fast-inhibitory processes that terminates individual puffs recovers quickly, a slower process terminates the flurry. This does not appear to result from IP$_3$Rs becoming inactivated by the rise of bulk cytosolic Ca$^{2+}$, because we found puff flurries were not suppressed by prior Ca$^{2+}$ elevations, and still terminated normally during global responses when cytosolic Ca$^{2+}$ levels were attenuated by buffering with EGTA (*Figure 4C,D*). Instead, our observations that partial depletion of ER Ca$^{2+}$ stores suppressed puff activity during global Ca$^{2+}$ responses (*Figure 5*) implicate the depletion of luminal Ca$^{2+}$ as a dominant mechanism responsible for terminating the local signals; analogous to the central role of luminal Ca$^{2+}$ depletion in terminating the Ca$^{2+}$ sparks mediated by ryanodine receptors (*Stern et al., 2013*). Although individual puffs appear not to affect luminal ER [Ca$^{2+}$] (*Lock et al., 2018*), it is plausible that ER depletion may occur during the rapid flurries of puffs evoked at multiple sites during the rising phase of global signals. A related question is whether the Ca$^{2+}$ depletion is locally confined to the ER around puff sites and arises through Ca$^{2+}$ released by the puffs themselves, or whether diffuse Ca$^{2+}$ liberation causes global ER Ca$^{2+}$ depletion throughout a luminally continuous ER network (*Okubo et al., 2015*; *Park et al., 2000*).

## Two functionally distinct modes of IP$_3$R-mediated Ca$^{2+}$ liberation

The two modes of Ca$^{2+}$ liberation we describe – punctate and diffuse – might, in principle, arise from two functionally and physically distinct populations of IP$_3$Rs, or through functional regulation of the clustered IP$_3$Rs underlying puffs such that they switch from a pulsatile to continuous mode of release. At present, we cannot discriminate between these mechanisms, but favor the former for congruence with studies revealing two distinct populations of IP$_3$Rs in terms of their spatial distributions and motilities. A small (~30%) fraction of the IP$_3$Rs in a cell are stationary (*Thillaiappan et al., 2017*; *Smith et al., 2014*), grouped in small clusters that are anchored at fixed sites predominantly near the plasma membrane (*Smith et al., 2009*; *Lock et al., 2018*; *Thillaiappan et al., 2017*), whereas the majority of IP$_3$Rs are distributed throughout the bulk of the cytoplasm and are motile within the ER membrane (*Thillaiappan et al., 2017*; *Smith et al., 2014*; *Tateishi et al., 2005*; *Gibson and Ehrlich, 2008*; *Ferreri-Jacobia et al., 2005*; *Fukatsu et al., 2004*). Puffs are proposed to originate from IP$_3$Rs within the immotile clusters that are endowed with the ability to preferentially respond to low concentrations of IP$_3$ (*Smith et al., 2009*; *Thillaiappan et al., 2017*). In contrast, the widely distributed, motile IP$_3$Rs remain apparently silent under conditions when puffs are selectively activated by low concentrations of IP$_3$ or in the presence of slow cytosolic Ca$^{2+}$ buffers to inhibit global Ca$^{2+}$ elevations (*Dargan and Parker, 2003*; *Dargan et al., 2004*). The motile, distributed IP$_3$Rs would be an attractive candidate for the diffusive mode of Ca$^{2+}$ liberation.

Functional differences between putative populations of IP$_3$Rs mediating punctate and diffuse Ca$^{2+}$ release cannot be attributed to their being constituted of different isoforms of IP$_3$R, because all three isoforms mediate Ca$^{2+}$ puffs (*Mataragka and Taylor, 2018*; *Lock et al., 2018*), and we show here that cells exclusively expressing individual isoforms generate cell-wide Ca$^{2+}$ elevations involving both punctate and diffuse modes of Ca$^{2+}$ liberation. Instead, the functional properties of the IP$_3$Rs may be affected by factors including their location in the cell, their mutual proximity to enable interactions by CICR, and by their association with modulatory and anchoring proteins (*Prole and Taylor, 2016*; *Lock et al., 2019*; *Konieczny et al., 2012*).

# Consequences of bimodal Ca$^{2+}$ signals for downstream signaling

Stimulation of the IP$_3$/Ca$^{2+}$ signaling pathway by activation of cell-surface receptors evokes repetitive oscillations in cytosolic Ca$^{2+}$ in numerous cell types (*Thomas et al., 1996*). Signaling information is encoded in the amplitude and frequency of these Ca$^{2+}$ oscillations, which may have periods ranging widely from a few seconds to minutes. Classical studies illustrate how different frequencies of Ca$^{2+}$ oscillations activate distinct targets, including the selective activation of NFkB in Jurkat T cells by low frequencies (*Dolmetsch et al., 1998*), and the frequency-dependent control of gene expression in RBL mast cells (*Li et al., 1998*). However, signaling information is not restricted to frequency and amplitude components of bulk cytosolic Ca$^{2+}$ elevations, and numerous findings implicate a component of spatial Ca$^{2+}$ profiling (*Smedler and Uhlén, 2014*; *Dupont et al., 2011*; *Berridge et al., 2003*; *Thul et al., 2009*). Because of the restricted diffusion of free Ca$^{2+}$ ions in the cytosol (*Allbritton et al., 1992*; *Schwaller, 2010*), the specificity of downstream signaling by Ca$^{2+}$ liberated through IP$_3$Rs will be strongly influenced by the proximity of target proteins, as well as by their Ca$^{2+}$ binding kinetics (*Samanta and Parekh, 2017*; *Atakpa et al., 2018*; *Csordás et al., 2010*). The two modes of IP$_3$-evoked Ca$^{2+}$ liberation we describe are, therefore, likely to selectively activate different populations of effectors; those positioned close to the IP$_3$Rs at puff sites that experience brief, repetitive transients of high local [Ca$^{2+}$], and others that respond to a more sustained, spatially diffuse elevation of bulk cytosolic [Ca$^{2+}$].

Based on a close juxtaposition of stationary IP$_3$R clusters to ER-plasma membrane junctions where STIM and Orai interact to induce store-operated Ca$^{2+}$ entry (SOCE) (*Thillaiappan et al., 2017*; *Thillaiappan et al., 2019*), it has been proposed that local depletion of ER Ca$^{2+}$ content at puff sites might rapidly and selectively activate SOCE, without requiring substantial overall loss of the ER Ca$^{2+}$ that is necessary to sustain numerous ER functions (*Thillaiappan et al., 2017*; *Thillaiappan et al., 2019*; *Taylor and Machaca, 2019*). On the other hand, we previously reported puffs to be unaffected by removal of extracellular Ca$^{2+}$ (*Lock et al., 2018*), and we show that patterning of local puff activity during IP$_3$-evoked global Ca$^{2+}$ signals is unaffected by the presence or absence of Ca$^{2+}$ in the bath solution. Thus, influx of extracellular Ca$^{2+}$ does not appear to contribute acutely to the initial flurry of puffs or to the rapid rise in global Ca$^{2+}$, although the more prolonged decay phase of the Ca$^{2+}$ signal in Ca$^{2+}$-containing medium points to a slower activation of SOCE. The relative contributions of local puffs vs. diffuse Ca$^{2+}$ liberation in activating SOCE remain to be determined. Another example of proximate Ca$^{2+}$ signaling is the close tethering between ER and mitochondria (*Giorgi et al., 2009*) that underlies a rapid shuttling of Ca$^{2+}$ released through IP$_3$Rs to the mitochondrial matrix (*Csordás et al., 2010*; *Filadi and Pozzan, 2015*), such that Ca$^{2+}$ transients within a signaling microdomain, rather than the bulk cytosolic Ca$^{2+}$ signal, regulate mitochondrial bioenergetics and induction of autophagy (*Cárdenas et al., 2010*). A similar close coupling has recently been described between IP$_3$Rs and lysosomes (*Atakpa et al., 2018*). Our description of two modes of IP$_3$-mediated Ca$^{2+}$ liberation thus raises questions regarding their respective roles in downstream signaling. Is organellar Ca$^{2+}$ signaling via structurally defined nanodomains restricted to the predominantly peripheral ER contact sites where puffs originate, or might a separate population of IP$_3$Rs that mediate diffuse Ca$^{2+}$ liberation also transmit their signals via restricted domains?

# Materials and methods

## Cell culture and loading

HEK293 wild-type (WT) cells were kindly provided by Dr. David Yule (University of Rochester). An IP$_3$R null cell line (3KO; #EUR030) and cell lines natively expressing exclusively type 1 (IP$_3$R1; #EUR031), type 2 (IP$_3$R2; #EUR032) and type 3 (IP$_3$R3; #EUR033) isoforms generated from that parental WT HEK293 cell line by CRISPR/Cas9 genetic engineering in the Yule lab were purchased from Kerafast (Boston, MA). HEK cell lines were characterized as described in *Alzayady et al., 2016* and were certified mycoplasma free prior to distribution. HeLa cells (#CCL-2) purchased from ATCC (Manassas, VA) were characterized by STR profiling and certified free of mycoplasma prior to distribution. HEK WT, 3KO, and single IP$_3$R isoform-expressing cell lines were cultured in Eagle's Minimum Essential Medium (EMEM; ATCC #30–2003), and HeLa cells were cultured in Dulbecco's modified Eagle Medium (DMEM; #11965092) from Thermo Fisher Scientific (Waltham, MA). Both EMEM and DMEM were supplemented with 10% fetal bovine serum (#FB-11) from Omega Scientific

(Tarzana, CA). All cell lines were cultured in plastic 75 cm flasks and maintained at 37°C in a humidified incubator gassed with 95% air and 5% $CO_2$. For imaging, cells were collected using 0.25% Trypsin-EDTA (Thermo Fisher Scientific #25200–056) and grown on poly-D-lysine (Millipore Sigma #P0899; St. Louis, MO) coated (1 mg/ml) 35 mm glass bottom imaging dishes (#P35-1.5–14 C) from MatTek (Ashland, MA) or 12 mm glass coverslips for 2–3 days.

Immediately before imaging, cells were incubated with the membrane-permeant fluorescent $Ca^{2+}$ indicator Cal520/AM (5 µM; AAT Bioquest #21130; Sunnyvale, CA) for 1 hr at room temperature (RT) in a $Ca^{2+}$-containing HEPES buffered salt solution ($Ca^{2+}$-HBSS). Where indicated cells were additionally loaded with membrane permeant esters of either the caged $IP_3$ analogue ci-$IP_3$/PM [D-2,3,-O-Isopropylidene-6-O-(2-nitro-4,5 dimethoxy) benzyl-myo-Inositol 1,4,5,-trisphosphate Hexakis (propionoxymethyl) ester] (1 µM; SiChem #cag-iso-2-145-10; Bremen, Germany) or the caged $Ca^{2+}$ buffer NP-EGTA [o-nitrophenyl EGTA/AM] (200–500 nM; Thermo Fisher Scientific #N6803) in conjunction with Cal520. For the experiments in *Figure 4C,D* cells were additionally loaded with EGTA/AM (15 µM; Thermo Fisher Scientific #E1219) for 1 hr at RT in $Ca^{2+}$-HBSS to attenuate global $Ca^{2+}$ elevations. To address possible saturation of Cal520 at high $Ca^{2+}$ levels, cells were alternatively loaded with the lower affinity $Ca^{2+}$ indicator fluo8L/AM (5 µM, AAT Bioquest #21096) together with 1 µM ci-$IP_3$/PM for 1 hr at RT in $Ca^{2+}$-HBSS. Following incubation with AM esters, cells were incubated for 30 min at room temperature in $Ca^{2+}$-HBSS. Cal520/AM, ci-$IP_3$/PM, and NP-EGTA/AM, EGTA/AM, and fluo8L/AM were all solubilized with DMSO/20% pluronic F127 (Thermo Fisher Scientific #P3000MP).

FCCP (carbonyl cyanide p-(trifluoromethoxy) phenylhydrazone) and CPA (cyclopiazonic acid), were purchased from Millipore Sigma (#C2920 and #C1530, respectively), and solubilized in 100% ethanol. Carbachol (#C4832) and histamine (#H7125), also from Millipore Sigma, were reconstituted in deionized $H_2O$. $Ca^{2+}$-HBSS contained (in mM) 135 NaCl, 5.4 KCl, 2 $CaCl_2$, 1 $MgCl_2$, 10 HEPES, and 10 glucose (pH = 7.4). Nominal $Ca^{2+}$-free HBSS consisted of the same formulation as $Ca^{2+}$-HBSS except that $CaCl_2$ was omitted; for zero $Ca^{2+}$-HBSS, 300 µM EGTA was added to nominal $Ca^{2+}$-free HBSS. For lattice light-sheet imaging, the plasma membrane was stained by adding Cell Mask Deep Red (Thermo Fisher Scientific #C10046) to the bathing solution in the imaging chamber at 1/50,000 dilution.

## $Ca^{2+}$ imaging

Total internal reflection fluorescence (TIRF) imaging of $Ca^{2+}$ signals was accomplished using a home-built system, based around an Olympus (Center Valley, PA) IX50 microscope equipped with an Olympus 60X oil immersion TIRF objective (NA 1.45). Fluorescence images were acquired with an Evolve EMCCD camera (Photometrics; Tucson, AZ) with a bit depth of 16 bits, using $2 \times 2$ binning for a final field at the specimen of $128 \times 128$ binned pixels (one binned pixel = 0.53 µm) at a rate of ~125 frames $s^{-1}$. Image data were streamed to computer memory using Metamorph v7.7 (Molecular Devices; San Jose, CA) and stored on hard disk for offline analysis.

Lattice light-sheet imaging was performed using a home-built system as previously described (*Ellefsen and Parker, 2018*). Images were acquired with an Andor Zyla 4.2 sCMOS camera (Oxford Instruments; Abingdon, England) from a single, diagonal light-sheet slice ($512 \times 256$ pixels, one pixel = 0.11 µm) at 100 frames $s^{-1}$ and 16 bit depth. $Ca^{2+}$ signals were imaged in Cal520-loaded cells for several seconds following photorelease of i-$IP_3$ by a flash from a 405 nm laser diode, utilizing 473 nm laser fluorescence excitation and a 510–560 nm bandpass emission filter. A 562 nm laser and 590 nm long-pass filter were used to image the plasma membrane stained with Cell Mask Deep Red. Images were streamed to disk using MicroManager (Vale Lab UCSF; San Francisco, CA).

### Photo-uncaging and local application of agonist

Photorelease of i-$IP_3$ was evoked by UV light from a xenon arc lamp filtered through a 350–400 nm bandpass filter and introduced by a UV-reflecting dichroic in the light path to uniformly illuminate the field of view. The amount of i-$IP_3$ released was controlled by varying the flash duration, set by an electronically controlled shutter (UniBlitz; Rochester, NY). The same system was used for photolysis of NP-EGTA (i.e. caged $Ca^{2+}$). For the local delivery of solutions to cells during imaging, glass micropipettes were prepared from borosilicate glass capillary filaments (1.5 mm x 0.86 mm, O.D. x I.D.) using a micropipette puller (Sutter Instruments; Novato, CA) to produce tip diameters of ~1–2 µm.

Micropipettes were positioned above the cell understudy with local delivery controlled using a pneumatic picospritzer. The delivery of micropipette contents and the duration and intensity of the UV-flash were empirically adjusted to evoke rapid rises in whole-cell cytosolic $Ca^{2+}$ levels.

All imaging was performed while cells were bathed in HBSS containing 2 mM $Ca^{2+}$ or zero $Ca^{2+}$-HBSS containing 0.3 mM EGTA and no added $Ca^{2+}$.

### Image analysis

Image data imported in16 bit integer MetaMorph stk or multi-plane TIF format were processed using a script written in Flika (http://flika-org.github.io), a freely available open-source image processing and analysis software in the Python programming language (*Ellefsen et al., 2019*; *Ellefsen et al., 2014*). All internal processing and data output were performed using 64 bit floating-point arithmetic.

To analyze and derive movies representing the pixel-by-pixel standard deviation (SD) of *temporal* fluctuations in fluorescence of the $Ca^{2+}$ indicator dye we used a custom Flika script as described previously (*Ellefsen et al., 2019*). In brief, following subtraction of camera black offset level, the raw image stack from the camera was first spatially filtered by a Gaussian blur function with a standard deviation (sigma) of 2 pixels (~1 μm at the specimen). The python package (skimage) used to perform this function applies a two-dimensional Gaussian blur with a specified standard deviation (sigma) out to a range of 4 sigma. To attenuate high-frequency photon shot noise and slow changes in baseline fluorescence, it was then temporally filtered with a bandpass Butterworth filter with low and high cutoff frequencies of 3 and 20 Hz. A running variance of this temporally filtered movie was calculated, pixel by pixel, by subtracting the square of the mean from the mean of the square of a moving 20 frame (160 ms) boxcar window of the movie. The running standard deviation was calculated by taking the square root of the variance image stack to create a standard deviation (SD) stack. Finally, to remove the mean predicted photon shot noise which increases in linear proportion to the mean fluorescence intensity, the SD stack was corrected, pixel-by-pixel, by subtracting the square root of a running mean of the spatially filtered fluorescence movie multiplied by a scalar constant; derived as illustrated in *Figure 1—figure supplement 2*.

We also applied a second FLIKA script to analyze *spatial* fluctuations in $Ca^{2+}$ image stacks. For this, black-level subtracted image stacks were first temporally band-pass filtered as before, and then new image stacks were derived by taking the difference of weak (sigma two pixel) and stronger (sigma eight pixel) Gaussian blur functions. Essentially, this functioned as a spatial bandpass filter, attenuating high frequencies caused by pixel-to-pixel shot noise variations and low frequency variations resulting from spread of $Ca^{2+}$ waves across the cell, while retaining spatial frequencies corresponding to the spread of local $Ca^{2+}$ puffs. Next, for each frame, we calculated the spatial variance among pixels within an imaging field, took the square root to obtain a measure of the spatial SD signal and subtracted the predicted increase in spatial shot noise with increasing fluorescence. Finally, we derived traces showing the average spatial fluctuations across a region encompassing the entire cell within a moving boxcar window of 160 ms (*Figure 1—figure supplement 3*).

The scripts used to perform temporal and spatial fluctuation analysis are presented as *Supplementary file 1*.

## Acknowledgements

We thank Dr. Carley A Karsten for assisting in initial imaging experiments and Dr. Kyle L Ellefsen for help with software and image analysis.

## Additional information

### Funding

| Funder | Grant reference number | Author |
|---|---|---|
| National Institute of General Medical Sciences | R37 GM048071 | Ian Parker |

The funders had no role in study design, data collection and interpretation, or the decision to submit the work for publication.

## Author contributions
Jeffrey T Lock, Data curation, Formal analysis, Writing - original draft, Writing - review and editing; Ian Parker, Conceptualization, Data curation, Formal analysis, Funding acquisition, Writing - original draft, Writing - review and editing

## Author ORCIDs
Jeffrey T Lock (iD) https://orcid.org/0000-0003-1522-3189

## Decision letter and Author response
Decision letter https://doi.org/10.7554/eLife.55008.sa1
Author response https://doi.org/10.7554/eLife.55008.sa2

# Additional files

## Supplementary files
• Supplementary file 1. $Ca^{2+}$ Image Processing Routines.

• Transparent reporting form

## Data availability
Algorithms used to generate SD fluctuation images from Ca2+ image recordings are provided in the Supplementary file 1.

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
