## [Decision Letter]

**Acceptance summary:**

Parker's pioneering work established that the initial responses to low concentrations of IP3 comprise Ca^2+^ puffs, wherein a few IP3Rs within an immobile cluster almost simultaneously open to evoke a local and short-lived cytosolic Ca^2+^ signal. It has been widely assumed that Ca^2+^ released by these events recruits further Ca^2+^ puffs (by Ca^2+^-induced Ca^2+^ release) to give global Ca^2+^ signals. It has, however, been difficult to examine this hypothesis because as Ca^2+^ puffs become more frequent, it becomes impossible to resolve them from background signal. The present work applies an elegant noise analysis method (the principles of which were described in a recent short report from the group in Cell Calcium) to assess the subcellular heterogeneity of Ca^2+^ signals evoked by IP3. The results are both unexpected and important in that they suggest that most cytosolic Ca^2+^ signals are not associated with the noise expected from Ca^2+^ puffs. The results convincingly challenge prevailing dogma by suggesting that Ca^2+^ puffs contribute only a small fraction the cytosolic Ca^2+^ signals evoked by IP3, with the remainder likely arising from openings of single IP3Rs. The authors have done an excellent job of revising the manuscript to address the technical concerns of the reviewers, and overall this is a rigorous, well done and significant body of work that justifies publication in *eLife*.

**Decision letter after peer review:**

[Editors’ note: the authors submitted for reconsideration following the decision after peer review. What follows is the decision letter after the first round of review.]

Thank you for submitting your work entitled "Inositol trisphosphate mediated Ca2+ spikes arise through two temporally and spatially distinct modes of Ca2+ release" for consideration by *eLife*. Your article has been reviewed by three peer reviewers, and the evaluation has been overseen by a Reviewing Editor and a Senior Editor. The following individuals involved in review of your submission have agreed to reveal their identity: Colin W Taylor (Reviewer #1); Grant Hennig (Reviewer #3).

Our decision has been reached after consultation between the reviewers. Based on these discussions and the individual reviews below, we regret to inform you that your work will not be considered further for publication in *eLife*.

The reviewers agreed that your work is interesting and has the potential to be quite important, but there were major concerns with the SD approach, on which most of the presented findings hinge, involving the dynamic range of the Ca^2+^ indicators and the spatial and temporal Ca^2+^ release patterns.

The first concern is that after uncaging or after the addition of agonists, the cytoplasmic background fluorescence in cells increases to a seemingly very high level. As punctate release events are only observed when background levels are low to moderate, it brings into question whether the Ca^2+^ indicators have become saturated at peak cytoplasmic background Ca^2+^ levels (the author's have previously assessed a variety of Ca^2+^ indicators with Kd values in the 150-400nM range). This would make it essentially impossible to resolve punctate events, even though punctate release events may still be occurring – akin to "signal clipping". While SD analysis may provide some additional resolution in situations with narrow dynamic fluorescence ranges, it still depends on the indicator having some dynamic range. This concern needs to be carefully addressed so that readers are confident that the presented findings are accurate and biological.

Some ways that could be used to examine the potential dynamic range issue of the indicators include:

• use a ratiometric indicator (Fura) to estimate actual cytoplasmic Ca^2+^ levels during uncaging/agonists which could be compared to Kd/saturation mM of Cal-520 and GCamP etc.

• pharmacologically limit the increase in cytoplasmic Ca^2+^ using low/moderate levels of EGTA (to preserve the dynamic range of the indicator) and examine whether punctate release patterns are still reduced after uncaging etc.

• purposefully increase cytoplasmic Ca^2+^ levels to near maximum with ionomycin then initiate an uncaging event to see puncate release sites can still be resolved (temporally synchronized release).

• use quantitative shape analysis (not ROIs) of transient bright regions at high cytoplasmic Ca^2+^ concentrations. At these levels where there is extremely limited dynamic range and greater shot noise, noise aggregates and punctate release sites will appear similar – but they can be separated using perimeter:area ratios (high for noise aggregates), or Gaussian fitting (better for release sites). This will require no Gaussian filtering of the data and some form of deconvolution to reduce the smoothing of objects due to Z-smear (where appropriate).

The second concern is discussed adequately in the reviews below, but questions whether the SD approach can detect synchronous, widespread puffs/ release events, or high frequency puffs/release events that appear and behave like changes in cytoplasmic background. It would be useful to the reader to see how different duration timespans affect the variance signal (which is currently tailored for the timecourse of individual puffs), to assess what range of Ca^2+^ behaviors are included or filtered by varying the space/time variance parameters.

Without additional information to address the major concerns of the reviewers, it is not possible to assess whether the work represents a sufficient advance to warrant publication in *eLife*.

Reviewer #1:

Parker's pioneering work established that the initial responses to low concentrations of IP3 comprise Ca^2+^ puffs, wherein a few IP3Rs within an immobile cluster almost simultaneously open to evoke a local and short-lived cytosolic Ca^2+^ signal. It has been widely assumed that Ca^2+^ released by these events recruits further Ca^2+^ puffs (by Ca^2+^-induced Ca^2+^ release) to give global Ca^2+^ signals. It has, however, been difficult to examine this hypothesis because as Ca^2+^ puffs become more frequent, it becomes impossible to resolve them from background signal. The present work applies an elegant noise analysis method (the principles of which were described in a recent short report from the group in Cell Calcium) to assess the subcellular heterogeneity of Ca^2+^ signals evoked by IP3. The results are both unexpected and important in that they suggest that most cytosolic Ca^2+^ signals are not associated with the noise expected from Ca^2+^ puffs. The results convincingly challenge prevailing dogma by suggesting that Ca^2+^ puffs contribute only a small fraction the cytosolic Ca^2+^ signals evoked by IP3, with the remainder likely arising from openings of single IP3Rs. The rigour and significance of the work justify publication in *eLife*; none of my comments should detract from that recommendation.

1) Figure 1 provides compelling evidence that noise decreases as the IP3-evoked Ca^2+^ signals develop and that a comparable increase in global cytosolic [Ca^2+^] evoked by ionomycin is “noise-free”, as expected for homogenous Ca^2+^ release. My understanding is that after some filtering and spatial blurring, most analyses calculate the SD for each pixel as a 160-ms (ie 20 frames) boxcar average. The original view posits that a global Ca^2+^ signal reflects most Ca^2+^ puff sites firing very frequently (perhaps 20 or so sites active at each instant), each spreading a µm or more from its source. Might this cause elevations in cytosolic [Ca^2+^] that are less noisy than the infrequent Ca^2+^ puffs observed initially, because each pixel would be influenced by Ca^2+^ reaching it from several puffs? It's perhaps too tall an order to require quantitative simulation of this on SD measurements, but it would be helpful to have the authors consider the issue explicitly.

2) From results in Figure 5, the authors propose that loss of Ca^2+^ from the ER causes puffs to terminate without appreciably abrogating the global increase in cytosolic [Ca^2+^]. That proposal would align with a long-standing, but contentious, suggestion that luminal Ca^2+^ supports IP3R activity, but it seems difficult to align with other work from this lab:

A paper from this lab and under review with *eLife* (Mak et al) proposes that luminal Ca^2+^ inhibits IP3R activity, such that loss of ER Ca^2+^ would be expected to enhance Ca^2+^ puffs.

Substantial evidence, much of it form this lab, indicates that individual Ca^2+^ puffs do not detectably affect ER luminal [Ca^2+^], since steps during the falling phase of a Ca^2+^ puff (reflecting channel closures) are of fixed amplitude. It is not thereby clear that the flurry of puff activity would cause the loss of ER Ca^2+^ proposed to terminate Ca^2+^ puffs.

The authors should discuss both issues.

Reviewer #2:

In this manuscript, Lock and Parker expand the ongoing work in the Parker lab to examine the temporal and spatial properties and their regulation of Ca^2+^ release by the inositol trisphosphate receptor (IP3R) Ca^2+^ release channel. They use HEK and HeLa cells to examine by TIRF microscopy (the primary approach here, and previously used extensively by this group) and light sheet microscopy, and a heavy dose of image processing abd fluctuation aanlyses, to address the issue of the sources of Ca^2+^ release and their mechanisms of transitioning from discrete Ca^2+^ signals, referred to as puffs, to global ones.

They find that IP3-mediated Ca^2+^ release is associated with a flurry of noisy events during the rising phase of the global Ca^2+^ signal that terminates before the peak of the global Ca^2+^ signal is achieved. They suggest that the noisy events are Ca^2+^-release events from IP3R clusters, i.e. Ca^2+^ puffs, on a background of IP3R-mediated more diffuse release. Using procedures to partially deplete Ca^2+^ stores, they conclude also that the Ca^2+^ puffs terminate because of depletion of the ER stores, although they cannot determine whether it represents local or global store depletion, whereas the diffuse release, from what they consider to be a separate population of IP3R from those in the Ca^2+^-puff inducing clusters, continue to release Ca^2+^. By measuring the rate of cytoplasmic Ca^2+^ concentration relaxation following a pulse of cytoplasmic Ca^2+^, they calculate the relative contributions of Ca^2+^ puffs (punctate release) and diffuse release during the rising phase and during the entire Ca^2+^ transient.

This is an elegant analysis and an interesting set of studies, and an not least of all because they address and challenge a major model of IP3-mediated local and global Ca^2+^ signaling (mostly developed by Parker); and because they address a conundrum in the field regarding the role and existence of "silent" IP3Rs that are localized throughout the cell but were previously (Parker; Taylor) considered to not participate. Here, the suggestion is that in addition to active IP3R in clusters near the plasma-membrane, these other IP3R channels might indeed be active and contribute to the diffuse release that continues after puffs have stopped firing.

I have a few comments that might suggest that the authors could clarify a few points.

1) Why aren't local release events observed in the raw ratio images, as Parker has described many times?

2) Figure 3—figure supplement 1 shows the sites with high standard deviation (SD) to be rather large, ~10 uM, a size that is inconsistent with the size of clusters of ~15 IP3R that generate the Ca^2+^ puffs. Rather their size suggests to me that these are entire regions of the endoplasmic reticulum (ER) that are near the plasma membrane in the TIRF evanescent field. If this is the case, are the authors selecting small sites within the large "contact" sites for the SD analyses? And, I would think that there is a "global"/homogenous signal at these sites that is contributed by Ca^2+^ that has diffused into these "contact" sites. This might also be consistent with the observation that the TIRF-detected high-SD signal terminates before the peak signal is achieved, because the peak is contributed by slow Ca^2+^ diffusion that persists after release has terminated.

3) The authors describe an inverted U relationship between the amplitude of the SD signal and the amplitude of the global response. What does it mean?

4) The authors conclude that elevated cytoplasmic Ca^2+^ doesn't inhibit the IP3R as a primary mechanism to terminate Ca puffs. Why doesn't it inhibit?

5) The authors examine the decay in cytoplasmic Ca^2+^ concentration after an elevation. They refer to this as sequestration. I'm not sure of their meaning. Although they suggest that the kinetics can be described by a single exponential, there must be at least two processes involved: sequestration by the ER, and extrusion by the plasm membrane Ca^2+^ pump. Indeed, in the discussion they use the phrase "efflux rate from the cytoplasm". I think what the authors mean to suggest in the sequestration away from the cytoplasm. If so, this should be stated more clearly.

Reviewer #3:

In this manuscript, the authors present a method that enables some resolution and measurement of small, punctate Ca^2+^ events in the context of global cytoplasmic Ca^2+^ levels. While previous studies used pharmacological tools to largely prevent global changes in Ca^2+^ in order to better reveal small discrete Ca^2+^ release events, the current study uses a variety of software routines that utilize fluorescence variance changes – from which punctate and global Ca^2+^ event behaviors are derived from. The authors surmise that punctate and global Ca^2+^ events are likely produced by 2 separate populations of IP3Rs. The experimental manipulations to investigate depletion of stores, Ca^2+^ entry, IP3R channel KO etc are well thought out and appropriate – but some concerns remain as to the ability of the authors to resolve puffs once background/cytoplasmic Ca^2+^ levels reach a certain intensity.

The use of signal variance to isolate different components of mixed fluorescence signals relies on those components having distinct (and variable) spatio-temporal signatures. However, if Ca^2+^ release events reach a steady state, then the variance technique becomes essentially useless. This is my main concern with some of the conclusions drawn from the analysis.

The ability to detect puffs is greatest during low background conditions, but the as global Ca^2+^ levels increase, the ability to detect puffs is hampered by i) closing of the max difference between background and puff sites and ii) an increase in the amplitude of shot noise that may start to mimick fast and small puff events. To resolve between genuine puff events, that are inherently Gaussian (1-2 µm) and random noise aggregates that occur during high background levels, is difficult – but is made worse by applying Gaussian filters to the data that reshape noise aggregates into more Gaussian shapes. Employing shape analysis before smoothing (perimeter:area ratios) is an effective way of resolving puffs (low P:A – smooth circles) even when surrounded by noise aggregates of equal intensity and size (high P:A – leafy looking). This may assuage some doubts as to the authors ability to resolve punctate events once background/global levels have reached moderate to high levels.

Similarly, the use of user-placed (or randomly positioned) ROIs based on visual inspection of bright spots in the videos is subjective and can heavily bias results to what the user wants to show. Many "hot spots" don't appear to have ROIs drawn over them and some ROIs appear to be sampling the same site. Small ROIs are more sensitive to spatial shifts in the position of the underlying fluorescent sites. The "flickering" variance may be a function of out-of-sync release events from a number of channels/clusters covered by each ROI. When these channels syncronize (in the open position?) or fluorescence sites increase in size, this will dramatically reduce the signal variance, even though puffs may still be occurring.

The shape of the cell and the distance and size of the cytoplasm located away from the evanescent field can dramatically affect the blurring of the background (cytoplasmic signal). How do the authors discriminate between "global" cytoplasmic changes and large puff sites that are heavily blurred being out of focus (see Figures 1 and 7). This effect still remains using lightsheet imaging.

The pixel resolution used during TIRF imaging was 0.5µm and is unlikely to fully resolve individual puffs, especially after smoothing. This may bias analysis to release events that are comprised of multiple channel clusters.

The use of black level subtraction can normalize absolute intensity levels, but does not compensate for the higher shot noise in areas with high static background. Was this taken into account?

Gaussian blur is reported as 2 pixels (in text) but as an SD of 2 pixels in the Materials and methods. What was the actual kernel size in pixels (3x3, 5x5?) and the SD.

Random ROIs were used for HEK cells devoid of IP3Rs. What does the data look like if the authors select the brightest spots in the 3KO video – or use random ROIs in controls?

Figure 3—figure supplement 3C appears to show a stochastic build up of high frequency events correlated to baseline increases – until background fluorescence levels may start to swamp out the ability to resolve small punctate events.

[Editors’ note: further revisions were suggested prior to acceptance, as described below.]

Thank you for choosing to send your work entitled "Inositol trisphosphate mediated Ca2+ spikes arise through two temporally and spatially distinct modes of Ca2+ release" for consideration at *eLife*. Your letter of appeal has been considered by a Senior Editor and the reviewers, and we are prepared to consider a revised submission.

The reviewers have also provided additional feedback below after reading your rebuttal to hopefully better convey the technical concerns.

Specific points:

1) Indicator Saturation/Clipping concerns:

In your rebuttal, you state (i) that "…maximal signal evoked by ionomycin is>2 fold higher than peak IP3-evoked fluorescence signals"

However, in Figure 1, the maximum raw signal during uncaging (1C) and after ionomycin (1F) looks exactly the same – both topping out at 8500 a.u – and suggests that IP3-stimulated Ca^2+^ release may be reaching levels that saturate the dye. Similarly, the large areas that show uniform white intensity (Figure 1, image v) are either due to: i) over zealous contrast/brightness adjustment, ii) indicator saturation @ max or iii) camera chip saturation (clipping). Assuming reasonable contrast adjustment and digitizing in 16-bits? (please add digitizer bit depth in Materials and methods), this suggests that saturation is still occurring. As none of the results figures presented data in absolute intensity values and used relative a.u. or F/F0 ratios which can give different "max levels" depending on the F0, it is hard for a reviewer to determine what the actual peak/max levels are and if indicator saturation was reached.

The importance of indicator saturation and/or Ca^2+^ release uniformity is magnified in using variance as the primary measure to resolve Ca^2+^ release events. Large, saturated areas have essentially no variance (after subtracting shot noise), and could dramatically underestimate SD values and obscure Ca^2+^ events. In a similar fashion, simultaneous firing of multiple puffs throughout a cell could also be difficult to interpret, as one would lose the ability to spatially discriminate individual events. This, also, would reduce SD values and make it hard to distinguish between cytoplasmic and "punctate" events.

While the use of Fluo8L, which has a much larger dynamic range in response to Ca^2+^ concentrations goes some way to assuage concerns over indicator saturation, Ca^2+^ levels may still be saturating this dye given the nature of the abrupt IP3-induced Ca^2+^ release. Out of curiosity, what does an IP3-uncaging event look like in a cell treated with ionomycin?

If the saturated/clipped regions from your recordings were dynamically filtered out using image processing and SD/intensity recalculated in parts of the cell in which the indicator is presumably unsaturated, this would help to allay saturation concerns on SD measurements.

2) Figure 1, ROIs

Towards the end of the rebuttal ("Summarizing key points…"), where you indicate the attention drawn to subjectively-placed ROIs was based on a "misconception", then one has to question ROIs were used at all. ROIs do illustrate important features of the Ca^2+^ release events, such as an apparent reduction in intensity variability in traces that reach near-maximal intensity (see first point).

3) 2 populations of IP3Rs…

Concerning point v) Figure 4 (we think you meant Figure 6), if potential methodological concerns are addressed – and indeed the punctate responses do drop out as global levels rise – the concluding statement that: "Our findings of a diffusive mode of Ca^2+^ liberation implicate a second population of IP3Rs with properties distinct from those clustered at puff sites" is too strong a statement given the evidence presented.

We think there needs to be experiments/analysis to show that punctate and global Ca^2+^ responses can be unrelated for this to be the case. In most examples presented the pattern of punctate release events appear to correlate with changes in cytoplasmic/global Ca^2+^.

For example, in Figure 2, the rate and amplitude of the SD events appears to be mirrored in the shape of the global responses. In E (puce color), the abrupt increase in SD is reflected as a sharp transition in global Ca^2+^ from baseline. In F (blue) the ramping up of SD events is reflected as a more gradual, sloping increase in global Ca^2+^. These relationships are illustrated nicely in G.

The maximum slope of global Ca^2+^ increases seems to occur at the point of maximum SD events. See Figure 4, Figure 5G and H (albeit de-amplified), Figure 6, Figure 8 (A: see 2 peaks in SD reflected in 2 steps in rate of Ca^2+^ intensity, like in Figure 3—figure supplement 3).

To disprove that there is just one population of IP3Rs, comparisons of the rate change of SD and global fluorescence should show little (or highly variable) correlation strength. Cumulative integration of SD and cumulative global fluorescence signals (akin to Ca^2+^ mass / SD mass) followed by regression, cross-correlation (with ∆ time offsets) and/or inflection analysis would be appropriate to examine those potential relationships. Performing this analysis on specific regions in cells to elucidate any potential zone-of-influence between punctate and global Ca^2+^ characteristics.

---

## [Author Response]

[Editors’ note: The authors appealed the original decision. What follows is the authors’ response to the first round of review.]

The reviewers agreed that your work is interesting and has the potential to be quite important, but there were major concerns with the SD approach, on which most of the presented findings hinge, involving the dynamic range of the Ca^2+^ indicators and the spatial and temporal Ca^2+^ release patterns.The first concern is that after uncaging or after the addition of agonists, the cytoplasmic background fluorescence in cells increases to a seemingly very high level. As punctate release events are only observed when background levels are low to moderate, it brings into question whether the Ca^2+^ indicators have become saturated at peak cytoplasmic background Ca^2+^ levels (the author's have previously assessed a variety of Ca^2+^ indicators with Kd values in the 150-400nM range). This would make it essentially impossible to resolve punctate events, even though punctate release events may still be occurring – akin to "signal clipping". While SD analysis may provide some additional resolution in situations with narrow dynamic fluorescence ranges, it still depends on the indicator having some dynamic range. This concern needs to be carefully addressed so that readers are confident that the presented findings are accurate and biological.

We agree that this is a crucial issue. Our original submission already included several lines of evidence indicating that fluorescence signals were well below the saturating level, even at the peak of large IP_3_-evoked responses. However, these data may not have been presented with sufficient clarity. We describe below our arguments and call out specific figures and places in the text where they are presented in our revised manuscript.

i) The maximal, saturating signals evoked by ionomycin were substantially higher than peak IP_3_-evoked fluorescence signals, and suppression of puff activity occurred at yet lower levels (paragraph two of subsection “Ca^2+^ puff activity terminates during the rising phase of global Ca^2+^ signals”). See also the sample traces we show in Author response image 1.

ii) We observed the same patterns of Ca^2+^ liberation using the low affinity indicator fluo-8L (Kd 1.86 µM) as compared to Cal520 (Kd 320 nM) which we used for the majority of our experiments (Figure 3—figure supplement 2).

iii) We were able to resolve instances of local puff activity even at the peak of IP_3_-evoked global fluorescence increases (Figure 3—figure supplement 1).

iv) Puff activity (SD signal) in cells exclusively expressing type 1 IP_3_R was suppressed during the rising phase of global Ca^2+^ signals and followed a similar pattern to WT cells, even though the magnitude of the peak fluorescence increase was <40% of that in WT cells (Figure 6).

v) The onset and termination of puff activity during the rise of IP_3_-mediated global Ca^2+^ signals were little altered when the initial basal Ca^2+^ level was elevated (Figure 4A,B) or, conversely, when the global Ca^2+^ rise was attenuated by buffering with cytosolic EGTA (new Figure 4C,D).

vi) The suppression of punctate Ca^2+^ liberation throughout all phases of IP_3_-evoked Ca^2+^ responses evoked when ER Ca^2+^ stores are partially depleted (Figure 5) strongly supports our proposal that Ca^2+^ liberation can arise in a diffuse manner that sustains Ca^2+^ signals after puff activity terminates during the rising phase of the response.

We thus conclude that the termination of puff activity during IP_3_-evoked signals reflects the physiological functioning of IP_3_Rs and is not an artifact of indicator saturation.

Some ways that could be used to examine the potential dynamic range issue of the indicators include:• use a ratiometric indicator (Fura) to estimate actual cytoplasmic Ca^2+^ levels during uncaging/agonists which could be compared to Kd/saturation mM of Cal-520 and GCamP etc.

We do not feel this approach is appropriate, with problems including uncertainties in determining the K_d_ of both Fura-2 and Cal520 in the cytosolic environment. Further, accurate calibration of Fura-2 signals could be impaired because Fura-2 itself may be nearly saturated; especially given that the affinity of Fura-2 is higher than Cal520. Instead, we believe our approach of comparing the peak amplitude of IP_3_-evoked Cal520 signals with the maximal (saturating) amplitude evoked by ionomycin provides a more direct and convincing evidence that IP_3_-evoked Cal520 signals were well below saturation.

• pharmacologically limit the increase in cytoplasmic Ca^2+^ using low/moderate levels of EGTA (to preserve the dynamic range of the indicator) and examine whether punctate release patterns are still reduced after uncaging etc.

This is a good suggestion. Thank you! We performed this experiment (luckily before our lab was shut down in response to the Covid19 crisis!) and incorporate the new data as Figure 4C,D. Our results show the same pattern of SD activity in EGTA loaded cells as in non-loaded cells – i.e. puff activity ceased before the peak of the global IP_3_-evoked Cal520 signal even though the peak amplitude of the global signal was much (~40%) lower than in control, non-EGTA-loaded cells (paragraph two subsection “Ca^2+^ puffs do not terminate because of rising cytosolic Ca^2+^ during cell-wide elevations.”).

• purposefully increase cytoplasmic Ca^2+^ levels to near maximum with ionomycin then initiate an uncaging event to see puncate release sites can still be resolved (temporally synchronized release).

Although performed for a different purpose, our experiments (Figure 4A,B) using photorelease from caged Ca^2+^ (rather than ionomycin) to elevate cytosolic [Ca^2+^] prior to evoking an IP_3_-mediated response already address this issue. Our results showed that puff activity was still triggered even with prior [Ca^2+^] elevations to levels (Cal520 DF/F_o_ of 6-8) at which puffs are suppressed during the rising phase of IP_3_-evoked responses beginning from baseline cytosolic [Ca^2+^].

• use quantitative shape analysis (not ROIs) of transient bright regions at high cytoplasmic Ca^2+^ concentrations. At these levels where there is extremely limited dynamic range and greater shot noise, noise aggregates and punctate release sites will appear similar – but they can be separated using perimeter:area ratios (high for noise aggregates), or Gaussian fitting (better for release sites). This will require no Gaussian filtering of the data and some form of deconvolution to reduce the smoothing of objects due to Z-smear (where appropriate).

This is an interesting suggestion, but developing, verifying and applying appropriate algorithms and software would incur a substantial time delay. We further contest the statement that dynamic range would be extremely limited. Mean peak Cal520 fluorescence signals evoked by i-IP_3_ were <40% of the maximal response evoked by ionomycin (paragraph two subsection “Ca^2+^ puff activity terminates during the rising phase of global Ca^2+^ signals”).

The second concern is discussed adequately in the reviews below, but questions whether the SD approach can detect synchronous, widespread puffs/ release events, or high frequency puffs/release events that appear and behave like changes in cytoplasmic background. It would be useful to the reader to see how different duration timespans affect the variance signal (which is currently tailored for the timecourse of individual puffs), to assess what range of Ca^2+^ behaviors are included or filtered by varying the space/time variance parameters.

We now include (new Figure 1—figure supplement 1) an evaluation of how varying the space/time variance parameters of the algorithm affect the resulting SD fluctuation signal. Other concerns are dealt with below in our responses to specific comments by the reviewers.

Without additional information to address the major concerns of the reviewers, it is not possible to assess whether the work represents a sufficient advance to warrant publication in eLife.

A major concern of the reviewers was that the decline of the SD signal (termination of puff activity) during the rise of global Ca^2+^ elevations could be attributed to an artifact of indicator saturation. We feel that this issue, and other concerns of the reviewers, are fully addressed by our responses detailed here, and that our findings provide convincing support for our proposal of two different functional modes of IP_3_-mediated signaling involving both punctate and diffuse Ca^2+^ liberation. Our model represents a major revision of the widely accepted “fire-diffuse-fire” notion by which global Ca ^2+^ signals are generated by coordination of transient, punctate release events. This represents an important advance and re-evaluation of current dogma in the field of Ca^2+^ signaling that we hope would warrant publication in *eLife*.

Reviewer #1:1) Figure 1 provides compelling evidence that noise decreases as the IP3-evoked Ca^2+^ signals develop and that a comparable increase in global cytosolic [Ca^2+^] evoked by ionomycin is “noise-free”, as expected for homogenous Ca^2+^ release. My understanding is that after some filtering and spatial blurring, most analyses calculate the SD for each pixel as a 160-ms (ie 20 frames) boxcar average.

We now include data (Figure 1—figure supplement 1) illustrating how the SD signal is affected by varying parameters for spatial filtering and boxcar averaging.

The original view posits that a global Ca^2+^ signal reflects most Ca^2+^ puff sites firing very frequently (perhaps 20 or so sites active at each instant), each spreading a µm or more from its source. Might this cause elevations in cytosolic [Ca^2+^] that are less noisy than the infrequent Ca^2+^ puffs observed initially, because each pixel would be influenced by Ca^2+^ reaching it from several puffs? It's perhaps too tall an order to require quantitative simulation of this on SD measurements, but it would be helpful to have the authors consider the issue explicitly.

We interpret the essence of this concern as to whether the continued, “noise-free” rise in cytosolic Ca^2+^ after puff activity ceases might arise as Ca^2+^ released during puffs diffuses into spaces between puff sites – i.e. that all the Ca^2+^ released during global responses originally derives from puffs. Perhaps the strongest argument supporting a distinct, “diffuse” mode of release comes from our experiments with partial depletion of ER Ca^2+^ content (Figure 5), where we show Ca^2+^ responses that arise in the absence of detectable puff activity. Moreover, a simple calculation indicates that Ca^2+^ diffusion would be too fast to account for the continuing smooth rise in Ca^2+^. Assuming a value of D = 20 mm^2^ s^-1^ for the effective diffusion coefficient of Ca^2+^, the mean time t for a Ca^2+^ ion to diffuse midway between puff sites separated by 10 mm ( d =10/2 mm) would be about 300 ms (t = d^2^/4D). In contrast, global Ca^2+^ signals typically continued to rise and were maintained for several seconds in the face of SERCA uptake after puff activity ceases (Figure 8). We had further addressed this issue in the context of whether Ca^2+^ diffusing from release sites deep in the cell interior might underlie the noise-free rise observed by TIRF imaging, but our lightsheet imaging indicated this was not the case (Figure 7).

We now explicitly discuss this issue in the revised manuscript (paragraph two subsection “Puff activity during global Ca^2+^ signals”).

2) From results in Figure 5, the authors propose that loss of Ca^2+^ from the ER causes puffs to terminate without appreciably abrogating the global increase in cytosolic [Ca^2+^]. That proposal would align with a long-standing, but contentious, suggestion that luminal Ca^2+^ supports IP3R activity, but it seems difficult to align with other work from this lab:A paper from this lab and under review with eLife (Mak et al) proposes that luminal Ca^2+^ inhibits IP3R activity, such that loss of ER Ca^2+^ would be expected to enhance Ca^2+^ puffs.

The proposal in the Mak et al. paper does not appear to be incompatible with our model. The proposed luminal regulation of IP_3_R activity would create a positive feedback loop, such that any local depletion of ER Ca^2+^ would speed Ca^2+^ liberation through puffs and more rapidly terminate that liberation if local stores at puff sites became more completely emptied. We prefer not to include discussion of this point at present, as the Mak et al. paper is not yet published.

Substantial evidence, much of it form this lab, indicates that individual Ca^2+^ puffs do not detectably affect ER luminal [Ca^2+^], since steps during the falling phase of a Ca^2+^ puff (reflecting channel closures) are of fixed amplitude. It is not thereby clear that the flurry of puff activity would cause the loss of ER Ca^2+^ proposed to terminate Ca^2+^ puffs.The authors should discuss both issues.

Findings of uniform step decrements of Ca^2+^ during the falling phase of puffs do suggest that there is little decline of ER Ca^2+^ during individual puffs evoked at relatively low frequency. However, it is plausible that ER depletion may occur during the rapid flurries of puffs evoked at multiple sites during the rising phase of global signals. We now incorporate discussion of this point (paragraph four subsection “Puff activity during global Ca^2+^ signals”).

Reviewer #2:[…]I have a few comments that might suggest that the authors could clarify a few points.1) Why aren't local release events observed in the raw ratio images, as Parker has described many times?

Local release events are indeed present in the raw ratio images. They are not readily visible in the figures because the grey scale range was adjusted to encompass the much higher fluorescence levels during global responses, and because puffs in the HEK cells we used for most experiments are smaller than in cell lines such as SHSY-5Y and HeLa. We now include a note in the text (Results paragraph two) to clarify the point for readers who may also have this question.

2) Figure 1—figure supplement 1 shows the sites with high standard deviation (SD) to be rather large, ~10 uM, a size that is inconsistent with the size of clusters of ~15 IP3R that generate the Ca^2+^ puffs. Rather their size suggests to me that these are entire regions of the endoplasmic reticulum (ER) that are near the plasma membrane in the TIRF evanescent field. If this is the case, are the authors selecting small sites within the large "contact" sites for the SD analyses? And, I would think that there is a "global"/homogenous signal at these sites that is contributed by Ca^2+^ that has diffused into these "contact" sites. This might also be consistent with the observation that the TIRF-detected high-SD signal terminates before the peak signal is achieved, because the peak is contributed by slow Ca^2+^ diffusion that persists after release has terminated.

All of our SD analyses (excepting Figure 1, where we show small regions of interest for purposes of illustrating the technique) were done using regions of interest encompassing the entire cell. There is thus no issue of “selecting small sites … for the SD analyses”. We now emphasize this point more clearly in the manuscript (subsection “Fluctuation analysis reveals a transient flurry of puffs during global Ca^2+^ signals”).

Please see our response to major point #1 of reviewer #1 regarding whether the peak is contributed by slow Ca^2+^ diffusion that persists after release has terminated.

3) The authors describe an inverted U relationship between the amplitude of the SD signal and the amplitude of the global response. What does it mean?

We show plots of amplitude of the SD signal and the amplitude of the global response as a convenient means of summarizing and depicting the relationship between these parameters in a time-independent manner. As described in the manuscript we originally thought the inverted U relationship might correspond to the “bell-shaped curve” of Ca^2+^ dependence of activation of IP_3_Rs. However, further experiments (Figure 4, now incorporating new data) indicated that is not the case. Our interpretation (Discussion, subsection “Puff activity during global Ca^2+^ signals”) is that the cytosolic Ca^2+^ signal is an inverse mirror of the local ER Ca^2+^ store level, and that the decline in SD signal arises as puff activity declines owing to falling ER store content.

4) The authors conclude that elevated cytoplasmic Ca^2+^ doesn't inhibit the IP3R as a primary mechanism to terminate Ca puffs. Why doesn't it inhibit?

Although high cytosolic [Ca^2+^] would be expected to inhibit opening of IP_3_R channels, a likely explanation is simply that depletion of ER Ca^2+^ stores is the more dominant mechanism to terminate puff activity during global signals. This interpretation is buttressed by new data we present (Figure 4C,D), demonstrating that puffs still terminate even when the global cytosolic Ca^2+^ elevation is substantially reduced by loading with EGTA.

5) The authors examine the decay in cytoplasmic Ca^2+^ concentration after an elevation. They refer to this as sequestration. I'm not sure of their meaning. Although they suggest that the kinetics can be described by a single exponential, there must be at least two processes involved: sequestration by the ER, and extrusion by the plasm membrane Ca^2+^ pump. Indeed, in the discussion they use the phrase "efflux rate from the cytoplasm". I think what the authors mean to suggest in the sequestration away from the cytoplasm. If so, this should be stated more clearly.

We agree that our use of the term “sequestration” was imprecise. We have replaced it by terms such as “removal from the cytosol”. Our experimental data fit well to a single exponential process, likely because sequestration by the ER is much faster than loss across the plasma membrane (e.g. Figure 5H)

Reviewer #3:[…] The experimental manipulations to investigate depletion of stores, Ca^2+^ entry, IP3R channel KO etc are well thought out and appropriate – but some concerns remain as to the ability of the authors to resolve puffs once background/cytoplasmic Ca^2+^ levels reach a certain intensity.The use of signal variance to isolate different components of mixed fluorescence signals relies on those components having distinct (and variable) spatio-temporal signatures. However, if Ca^2+^ release events reach a steady state, then the variance technique becomes essentially useless. This is my main concern with some of the conclusions drawn from the analysis.

By the statement “Ca^2+^ release events reach a steady state” we interpret the reviewer to mean that Ca^2+^ release through the IP_3_Rs that mediate puffs becomes continuous; i.e. that some or all of the IP_3_R channels remain open all the time, resulting in a “noise-free” signal. If this indeed occurs, we believe it is valid and useful to regard such continuous release as a mode distinct from the transient puffs. The issue, then, is whether the same IP_3_Rs underlie puffs and continuous release, or whether these modes arise from separate populations of IP_3_Rs. Although we favor the notion that distinct populations of IP_3_Rs mediate the two modes of Ca^2+^, we now explicitly discuss the possibility the two modes may instead arise through a switch of the same IP_3_Rs from one form of regulation to another (Discussion and see response to similar query of reviewer #1).

The ability to detect puffs is greatest during low background conditions, but the as global Ca^2+^ levels increase, the ability to detect puffs is hampered by i) closing of the max difference between background and puff sites and ii) an increase in the amplitude of shot noise that may start to mimick fast and small puff events. To resolve between genuine puff events, that are inherently Gaussian (1-2 µm) and random noise aggregates that occur during high background levels, is difficult – but is made worse by applying Gaussian filters to the data that reshape noise aggregates into more Gaussian shapes. Employing shape analysis before smoothing (perimeter:area ratios) is an effective way of resolving puffs (low P:A – smooth circles) even when surrounded by noise aggregates of equal intensity and size (high P:A – leafy looking). This may assuage some doubts as to the authors ability to resolve punctate events once background/global levels have reached moderate to high levels.

Use of shape analysis is an interesting suggestion; but developing, verifying and applying appropriate algorithms and software would incur a substantial time delay. We feel that the data we already present convincingly demonstrate that the fall in SD signal during the rising phase of global Ca^2+^ responses truly represents a decline in puff activity, not a failure to detect puffs in the face of elevated Ca^2+^levels. Notably; (i) Puff activity (SD signal) was still observed in response to IP_3_ activation when the basal cytosolic Ca^2+^ was substantially elevated by prior photorelease of Ca^2+^ (Figure 4A,B). (ii) We now include new data showing that puff termination follows similar kinetics when peak global Ca^2+^ elevations were reduced by EGTA buffering (Figure 4C,D). Moreover, we present additional evidence in our rebuttals to concerns of the other reviewers that our ability to detect puffs at elevated Ca^2+^ levels may have been impaired by limited dynamic range of the indicator.

Similarly, the use of user-placed (or randomly positioned) ROIs based on visual inspection of bright spots in the videos is subjective and can heavily bias results to what the user wants to show. Many "hot spots" don't appear to have ROIs drawn over them and some ROIs appear to be sampling the same site. Small ROIs are more sensitive to spatial shifts in the position of the underlying fluorescent sites. The "flickering" variance may be a function of out-of-sync release events from a number of channels/clusters covered by each ROI. When these channels syncronize (in the open position?) or fluorescence sites increase in size, this will dramatically reduce the signal variance, even though puffs may still be occurring.

We show SD traces from small, subcellular ROIs only in Figure 1, for the purpose of better illustrating the basis of our technique. For all the other experiments – from which our conclusions are drawn – we used a single ROI encompassing the entire cell. There is thus no issue of subjective bias resulting from user-placed ROIs. The reviewer appears to have missed this methodological point, which we now emphasize more clearly in the text (subsection “Fluctuation analysis reveals a transient flurry of puffs during global Ca^2+^ signals”).

Our SD signal represents a measure of temporal fluctuations aggregated across all pixels covering the cell. It will thus be substantially unaffected whether local puffs are synchronized or asynchronous. Of course, if channels all “synchronize” in a continuously open state there would be no fluctuations – but as described above, we would regard this as a distinctly different mode of release.

The shape of the cell and the distance and size of the cytoplasm located away from the evanescent field can dramatically affect the blurring of the background (cytoplasmic signal). How do the authors discriminate between "global" cytoplasmic changes and large puff sites that are heavily blurred being out of focus (see Figures 1 and 7). This effect still remains using lightsheet imaging.

Again, all our analyses are based on SD and DF/F0 traces derived from the same, single ROI encompassing each cell. They represent an average of the temporal fluctuations and Ca^2+^ level across the entire TIRF footprint of the cell. We do not attempt to discriminate on the basis of spatial patterns between global signals and large puffs.

The pixel resolution used during TIRF imaging was 0.5µm and is unlikely to fully resolve individual puffs, especially after smoothing. This may bias analysis to release events that are comprised of multiple channel clusters.

The camera pixel resolution, subsequent Gaussian smoothing, and boxcar calculation of SD were chosen to optimally “tune” for the spatio-temporal characteristic of puffs while minimizing stochastic photon shot noise. The SD signal simply reports an average of pixel-by-pixel temporal fluctuations across the entire cell. We do not attempt to discriminate between individual or multi-cluster events.

In response to the editor’s comment we now include a supplementary figure (Figure 1—figure supplement 1) illustrating how different parameters in our algorithm affect the SD signal. The spatial smoothing we applied (σ corresponding to ~1 mm at the specimen) effectively reduced photon shot noise see without smoothing, but did not otherwise appreciably alter the SD signal arising from puff activity.

The use of black level subtraction can normalize absolute intensity levels, but does not compensate for the higher shot noise in areas with high static background. Was this taken into account?

Our algorithm subtracts the mean shot noise level predicted for the intensity level on a pixel-by-pixel basis (subsection “Fluctuation processing of Ca^2+^ images highlights transient signals” paragraph three).

Gaussian blur is reported as 2 pixels (in text) but as an SD of 2 pixels in the Materials and methods. What was the actual kernel size in pixels (3x3, 5x5?) and the SD.

Spatial smoothing was accomplished using a Python routine (skimage) that applies a two-dimensional Gaussian blur with specified standard deviation (σ) out to a range of 4 σ. We now clarify this description, and state σ in terms of both pixels and corresponding micrometers at the specimen (subsection “Image Analysis”). See new Figure 1—figure supplement 1 for an illustration of how the SD signal is affected by different values of Gaussian blur.

Random ROIs were used for HEK cells devoid of IP3Rs. What does the data look like if the authors select the brightest spots in the 3KO video – or use random ROIs in controls?

As noted above, we show traces from small, subcellular ROIs only in Figure 1, for the purpose of helping explain the basis of our SD approach. Use of random ROIs for the 3KO cell would seem appropriate for that purpose. Excepting the introductory Figure 1, all data were acquired from single regions of interest encompassing the entire cell (yellow outlines and traces in Figure 1).

Figure 3—figure supplement 3C appears to show a stochastic build up of high frequency events correlated to baseline increases – until background fluorescence levels may start to swamp out the ability to resolve small punctate events.

As described in responses to several of the reviewers’ comments, we present convincing evidence that fluctuations from punctate events do not become “swamped out” at higher background fluorescence levels, but rather that the punctate release activity itself declines.

[Editors’ note: what follows is the authors’ response to the second round of review.]

The reviewers have also provided additional feedback below after reading your rebuttal to hopefully better convey the technical concerns.Specific points:1) Indicator Saturation/Clipping concerns:In your rebuttal, you state (i) that "…maximal signal evoked by ionomycin is>2 fold higher than peak IP3-evoked fluorescence signals"However, in Figure 1, the maximum raw signal during uncaging (1C) and after ionomycin (1F) looks exactly the same – both topping out at 8500 a.u – and suggests that IP3-stimulated Ca^2+^ release may be reaching levels that saturate the dye.

We apologize that we had not fully described the procedure used to evoke the ionomycin response in Figure. 1, leading to the reviewer’s confusion. The experiment in Figure 1F,H was done under conditions deliberately chosen to evoke a Ca^2+^ fluorescence signal matching the amplitude and kinetics of the IP_3_-evoked signal in Figure 1A,C. Our intent was to compare the output of our SD analysis routine with signals of similar size where punctate release was either present or absent. For this purpose, we delivered a small aliquot of ionomycin at a distance (determined by trial and error) from cells imaged in zero Ca^2+^ solution, so that the slow diffusion and dilution of ionomycin evoked gradual release of Ca^2+^ from intracellular stores.

Separately, we used ionomycin to determine the saturating fluorescence of the indicator by complete bath exchange of medium containing high (10 mM) calcium with ionomycin. We now clarify these points in the text, and also state the peak amplitudes of the ionomycin-and i-IP_3_-evoked evoked response in Figure 1 as ΔF/F0 values as well as in fluorescence units.

Similarly, the large areas that show uniform white intensity (Figure 1, image v) are either due to: i) over zealous contrast/brightness adjustment, ii) indicator saturation @ max or iii) camera chip saturation (clipping). Assuming reasonable contrast adjustment and digitizing in 16-bits? (please add digitizer bit depth in Materials and methods), this suggests that saturation is still occurring.

Camera digitization was 16 bit; the camera was operated far below full-well capacity; and image processing was done using 64 bit floating point arithmetic. No clipping occurred anywhere through the entire imaging pipeline. This is now stated in Materials and methods.

Areas of uniform white in Figure 1A,F resulted because we chose a grey scale that would still display some data at low fluorescence levels, and this inadvertently clipped the higher intensity values. We have now re-made this figure with a more extended grey scale so that the Cal520 fluorescence image panels at the peak of the response are no longer clipped.

As none of the results figures presented data in absolute intensity values and used relative a.u. or F/F0 ratios which can give different "max levels" depending on the F0, it is hard for a reviewer to determine what the actual peak/max levels are and if indicator saturation was reached.

The term “absolute intensity value” (even if expressed as photons/pixel/unit time) has little meaning in this context. The fluorescence intensity will depend on factors including excitation laser intensity, concentration of indicator loaded in the cytosol, TIRF angle setting, etc.; factors that will vary between cells and between experiments. Instead, we follow the widely used practice of expressing data as ΔF/F0 values to normalize for these factors. The only crucial underlying assumption is that the basal free Ca^2+^ level (setting F0) is consistent between cells prepared under the same conditions. From many years’ experience, including comparison measurements with fura-2, we find this to be the case.

For reference, we show in Author response image 1 representative traces of strong responses to i-IP_3_ in 3 cells black), and maximal responses to ionomycin in 3 cells (red). These data are displayed as both “raw” fluorescence in arbitrary camera units (left) and as F/F0 values (right).

The importance of indicator saturation and/or Ca^2+^ release uniformity is magnified in using variance as the primary measure to resolve Ca^2+^ release events. Large, saturated areas have essentially no variance (after subtracting shot noise), and could dramatically underestimate SD values and obscure Ca^2+^ events.

While we agree that our data and conclusions would be dramatically affected if the indicator became saturated, we present numerous lines of evidence (including new data – Figure 4C,D – incorporated in our revised manuscript) clearly demonstrating that IP_3_-evoked Ca^2+^ signals were well below the level of indicator saturation, and that our key conclusion that puff activity terminates before the peak of global signals is not an artifact of indicator saturation.

In a similar fashion, simultaneous firing of multiple puffs throughout a cell could also be difficult to interpret, as one would lose the ability to spatially discriminate individual events. This, also, would reduce SD values and make it hard to distinguish between cytoplasmic and "punctate" events.

Our SD signal reflects the cell-wide aggregate of fluctuations at each pixel, and is independent of any requirement to spatially discriminate individual events.

While the use of Fluo8L, which has a much larger dynamic range in response to Ca^2+^ concentrations goes some way to assuage concerns over indicator saturation, Ca^2+^ levels may still be saturating this dye given the nature of the abrupt IP3-induced Ca^2+^ release.

We do not understand this point. Ca^2+^ binding to the indicator (and its fluorescence) will equilibrate with free [Ca^2+^] on a timescale vastly more rapidly than the timescale of seconds for the IP_3_-evoked signal.

Out of curiosity, what does an IP3-uncaging event look like in a cell treated with ionomycin?

Because ionomycin will cause uncontrolled loss of Ca^2+^ from the ER, we would not expect any response to uncaged IP_3_ if no Ca^2+^ remained to be released. In Figure 4A,B we show i-IP_3_-evoked responses after cytosolic [Ca^2+^] was elevated using caged calcium.

If the saturated/clipped regions from your recordings were dynamically filtered out using image processing and SD/intensity recalculated in parts of the cell in which the indicator is presumably unsaturated, this would help to allay saturation concerns on SD measurements.

Again, we present compelling evidence that our IP_3_-evoked signals are well below the level of indicator saturation. Our revised manuscript includes new experimental data, and a new section in the Discussion summarizing and evaluating this evidence.

As shown in (revised) Figure 1A and Video 1, the fluorescence at the peak of IP_3_-evoked signals is quite homogeneous throughout the cell (TIRF footprint), without evidence of saturated/clipped regions.

2) Figure 1, ROIsTowards the end of the rebuttal ("Summarizing key points…"), where you indicate the attention drawn to subjectively-placed ROIs was based on a "misconception", then one has to question ROIs were used at all. ROIs do illustrate important features of the Ca^2+^ release events, such as an apparent reduction in intensity variability in traces that reach near-maximal intensity (see first point).

We show traces from subcellular ROIs in Figure 1 as a way to illustrate the behavior of puff sites (particularly to provide helpful background for readers who may not be familiar with this literature), and to show how the aggregate behavior of multiple sites results in the whole-cell ROI traces we present in the rest of the manuscript. We have reworded the text to better communicate this intent. We agree that there is valuable information to be derived from analysis of individual release events, but leave that analysis for a future publication.

3) 2 populations of IP3Rs…Concerning point v) Figure 4 (we think you meant Figure 6), if potential methodological concerns are addressed – and indeed the punctate responses do drop out as global levels rise – the concluding statement that: "Our findings of a diffusive mode of Ca^2+^ liberation implicate a second population of IP3Rs with properties distinct from those clustered at puff sites" is too strong a statement given the evidence presented.

We agree that our statement was too strong. In our revised manuscript we now refer to two “modes” of Ca^2+^ release, and in the Discussion speculate as to whether these modes may arise from two separate populations of IP_3_R, or through a modulation of the functional properties of a single population. We further address this point in our detailed responses to the original comments of the reviewers.

We think there needs to be experiments/analysis to show that punctate and global Ca^2+^ responses can be unrelated for this to be the case.

Figure 5A,B,D,F,H,I present clear-cut cases where a global Ca^2+^ signal is evoked without detectable puff activity (SD signal); i.e. in this situation (partial depletion of ER Ca^2+^) punctate and diffuse modes of Ca^2+^ release are indeed unrelated.

In most examples presented the pattern of punctate release events appear to correlate with changes in cytoplasmic/global Ca^2+^.For example, in Figure 2, the rate and amplitude of the SD events appears to be mirrored in the shape of the global responses. In E (puce color), the abrupt increase in SD is reflected as a sharp transition in global Ca^2+^ from baseline. In F (blue) the ramping up of SD events is reflected as a more gradual, sloping increase in global Ca^2+^. These relationships are illustrated nicely in G.The maximum slope of global Ca^2+^ increases seems to occur at the point of maximum SD events. See Figure 4, Figure 5G and H (albeit de-amplified), Figure 6, Figure 8 (A: see 2 peaks in SD reflected in 2 steps in rate of Ca^2+^ intensity, like in S5).

This is consistent with, and predicted by, our proposal that punctate Ca^2+^ release though puffs contributes a substantial (~40%; Figure 8A-C) fraction of the total Ca^2+^ liberation during the rising phase of global Ca^2+^ signals. We thus expect that the rising phase of the global signal will reflect both punctate and diffuse modes of Ca^2+^ liberation.

To disprove that there is just one population of IP3Rs, comparisons of the rate change of SD and global fluorescence should show little (or highly variable) correlation strength. Cumulative integration of SD and cumulative global fluorescence signals (akin to Ca^2+^ mass / SD mass) followed by regression, cross-correlation (with ∆ time offsets) and/or inflection analysis would be appropriate to examine those potential relationships.

As noted above, our revised manuscript refers to two modes of Ca^2+^ release, and we discuss whether these may arise from two populations of IP_3_Rs or a single population. Because we conclude that punctate Ca^2+^ release makes up an appreciable part of the total Ca^2+^ released during the rising phase of the cell-wide response, we do expect correlation between the SD signal and the global fluorescence.

Performing this analysis on specific regions in cells to elucidate any potential zone-of-influence between punctate and global Ca^2+^ characteristics.

This suggestion appears predicated on a proposal that punctate Ca^2+^ and diffuse Ca^2+^ release arise at different sites in the cell. An interesting question, but one we feel is beyond the scope of the present manuscript, and a topic for future experiments once our lab reopens after the Covid-19 pandemic.